# Insect detect: An open-source DIY camera trap for automated insect monitoring

**Maximilian Sittinger** [ID] *, **Johannes Uhler** [ID], **Maximilian Pink** [ID], **Annette Herz**

Julius Kühn Institute (JKI)—Federal Research Centre for Cultivated Plants, Institute for Biological Control, Dossenheim, Germany

* maximilian.sittinger@idiv.de

**Data Availability Statement:** The camera trap software and insect detection models are available at GitHub (https://github.com/maxsitt/insect-detect). The insect classification model, the Python script for metadata post-processing and model

## Abstract

Insect monitoring is essential to design effective conservation strategies, which are indispensable to mitigate worldwide declines and biodiversity loss. For this purpose, traditional monitoring methods are widely established and can provide data with a high taxonomic resolution. However, processing of captured insect samples is often time-consuming and expensive, which limits the number of potential replicates. Automated monitoring methods can facilitate data collection at a higher spatiotemporal resolution with a comparatively lower effort and cost. Here, we present the Insect Detect DIY (do-it-yourself) camera trap for non-invasive automated monitoring of flower-visiting insects, which is based on low-cost off-the-shelf hardware components combined with open-source software. Custom trained deep learning models detect and track insects landing on an artificial flower platform in real time on-device and subsequently classify the cropped detections on a local computer. Field deployment of the solar-powered camera trap confirmed its resistance to high temperatures and humidity, which enables autonomous deployment during a whole season. On-device detection and tracking can estimate insect activity/abundance after metadata post-processing. Our insect classification model achieved a high top-1 accuracy on the test dataset and generalized well on a real-world dataset with captured insect images. The camera trap design and open-source software are highly customizable and can be adapted to different use cases. With custom trained detection and classification models, as well as accessible software programming, many possible applications surpassing our proposed deployment method can be realized.

## Introduction

The worldwide decline in insect biomass, abundance and diversity has been reported in numerous studies in recent years [1–3]. To identify potential drivers of insect decline, quantify their impact and design effective conservation strategies, more long-term monitoring data across ecological gradients is needed, preferably at high spatiotemporal resolutions [4, 5]. An obstacle to the implementation of large-scale monitoring schemes are often financial restrictions. The estimated costs for a national pollinator monitoring scheme range from ~525 € per site and year for volunteer floral observations with crowd-sourced identification to group level, up to ~8130 € for a professional monitoring with pan traps, transect walks, floral

training notebooks are available at GitHub (https://github.com/maxsitt/insect-detect-ml). The source files for the documentation website are available at GitHub (https://github.com/maxsitt/insect-detect-docs). The modified YOLOv5 scripts, including for classification of insect images, are available at GitHub (https://github.com/maxsitt/yolov5). The datasets for insect detection model training (https://doi.org/10.5281/zenodo.7725941) and insect classification model training (https://doi.org/10.5281/zenodo.8325384) are available at Zenodo. The R scripts and associated data for creation of plots shown in this paper are available at Zenodo (https://doi.org/10.5281/zenodo.10171524).

**Funding:** The author(s) received no specific funding for this work.

**Competing interests:** The authors have declared that no competing interests exist.

observations and expert identification to species level [6]. Due to these high costs, as well as the time-consuming processing and identification of insect specimens, traditional monitoring methods are usually deployed with a low number of spatial and/or temporal replicates. This often generates snapshot data with restricted potential for analysis and interpretation.

Non-invasive automated monitoring methods, which can acquire data at a high spatiotemporal resolution, can complement traditional monitoring methods that often generate data with a comparatively higher taxonomic resolution. When integrated into frameworks based on artificial intelligence (AI) approaches for rapid automated data processing and information extraction, large amounts of high-quality data can be collected with comparatively lower time and effort required [7]. Standardizing these automated monitoring methods and providing easy accessibility and reproducibility could furthermore decentralize monitoring efforts and strengthen the integration of independent biodiversity observations, also by non-professionals (Citizen Science) [8].

A range of different sensors can be used for automated insect monitoring [9]. These include acoustic [10] and opto-electronic sensors [11–13], as well as cameras [14–20]. Several low-cost DIY camera trap systems use scheduled video or image recordings, which are analyzed in subsequent processing steps [14, 15]. Other systems utilize motion detection software as trigger for the video or image capture [16, 18]. Similar to traditional camera traps used for the monitoring of mammals, these systems often generate large amounts of video/image data, which is most efficiently processed and analyzed by making use of machine learning (ML) and especially deep learning (DL) algorithms [21–23] automatically extract information such as species identity, abundance or behavior [24]. While automated insect classification with specifically trained DL models does not yet reach the accuracy of taxonomic experts, results are still of sufficient quality to perform routine identification tasks in a fraction of time with the potential to significantly reduce human workload [25].

Small DL models with few parameters and relatively low computational costs can be run on suitable devices directly in the field ("AI at the edge"), to enable real-time detection and/or classification of objects the model was trained on. An existing camera trap for automated insect monitoring combines scheduled time-lapse image recordings at 0.33 fps (frames per second) with subsequent on-device insect detection and classification running in parallel, while tracking is implemented during post-processing on a local computer [17]. As an alternative approach, frames produced by the camera can also be used as direct input for an insect detection model running with a high frame rate in real time on the camera trap hardware. In this way, the appearance and detection of an insect can automatically trigger the image capture. If the model is optimized to detect a wide range of possibly occurring insects with a high accuracy, this is usually more robust to false triggers (e.g. moving background) compared to motion detection and can drastically reduce the amount of data that has to be stored. Smaller volumes of data subsequently enable faster and more efficient post-processing even on standard computer hardware. In contrast to systems utilizing cloud computing, there are no networking costs or dependence on wireless broadband coverage and lower power requirements. Furthermore, new possibilities arise with having more information immediately available on the device, especially regarding autonomous decision making (e.g. automatically adjust recording times to capture more insects) or sending small sized metadata with pre-processed information at the end of a recording interval.

Following this approach, we present the design and proof of concept of a novel DIY camera trap system for automated visual monitoring of flower-visiting insects. To the best of our knowledge, there is currently no system available that combines the real-time processing capabilities described in the following, with a completely open software environment, including simple no-code training of state-of-the-art detection and classification models that can be deployed and modified also by non-professionals. The system is based on low-cost off-the-

shelf hardware components and open-source software, capable of on-device AI inference at the edge with custom trained models. Our goal was to develop a camera trap that could be easily utilized in monitoring projects involving citizen scientists to achieve a broader application potential. As such, it should be inexpensive, easy to assemble and set up, and provide reliable results without the requirement of expert taxonomic knowledge during data collection. Detailed step-by-step instructions to enable a simple reproduction and optional customization of the camera trap can be found at the corresponding documentation website [26].

The camera trap is resistant to high temperatures and humidity, as well as fully solar-powered, which makes it possible to deploy autonomously during a whole season. Insects landing on a platform with colored flower shapes are detected and tracked in real time at ~12.5 fps, while an image of each detected insect is cropped from synchronized high-resolution frames (1920x1080 pixels) and saved together with relevant metadata to a microSD card every second. The insect images can then be automatically classified with a custom trained model in a subsequent step on a local computer. The on-device tracking capabilities were tested with a fast-flying hoverfly species (*Episyrphus balteatus*) in a lab experiment, and different metadata post-processing settings for activity/abundance estimation were compared. Five camera traps were continuously deployed during four months in 2023 with a total recording time of 1919 hours. Captured images and metadata were classified, post-processed and analyzed with a focus on six different hoverfly species(-groups). The generalization capability of the classification model was validated on a real-world dataset of images captured during field deployment.

## Materials and methods

### Hardware

The camera trap system is based on three main hardware components: (1) OpenCV AI Kit (OAK-1) (Luxonis, Littleton, USA), including a 12MP image sensor and a specific chip for on-device AI inference with custom models; (2) Raspberry Pi Zero 2 W (RPi) (Raspberry Pi Foundation, Cambridge, UK), a single-board computer that is used as host for the OAK-1 and stores the captured data on a microSD card; (3) PiJuice Zero pHAT (Pi Supply, East Sussex, UK), an expansion board for the RPi with integrated RTC (real-time clock) for power management and scheduling of recording times.

Two rechargeable batteries (~91Wh combined capacity) connected to a 9W 6V solar panel (Voltaic Systems, New York, USA) are used as power supply for the system. All electronic components are installed in a weatherproof enclosure, which can be mounted to a standard wooden or steel post (Fig 1B). A simple hardware schematic with more details about the connections between the individual components can be found in the supporting information (S1 Fig). Below the camera trap enclosure, a sheet (e.g. acrylic glass or lightweight foam board) with colored flower shapes printed on top is attached to the same post, which acts as visual attractant and landing platform for flower-visiting insects (Fig 1A). This standardizable platform (e.g. 50x28 or 35x20 cm) provides a homogeneous plane background and leads to a uniform posture of insects sitting on it, which increases detection, tracking and classification accuracy with less data required for model training. The cost for the full camera trap setup including all components is ~700 €, a minimal setup can be built for ~530 €. A complete list of all required components and step-by-step assembly instructions can be found at the documentation website [26].

### Software

All of the camera trap software and associated insect detection models are available at GitHub [27]. The software includes Python scripts for livestreaming of camera frames together with

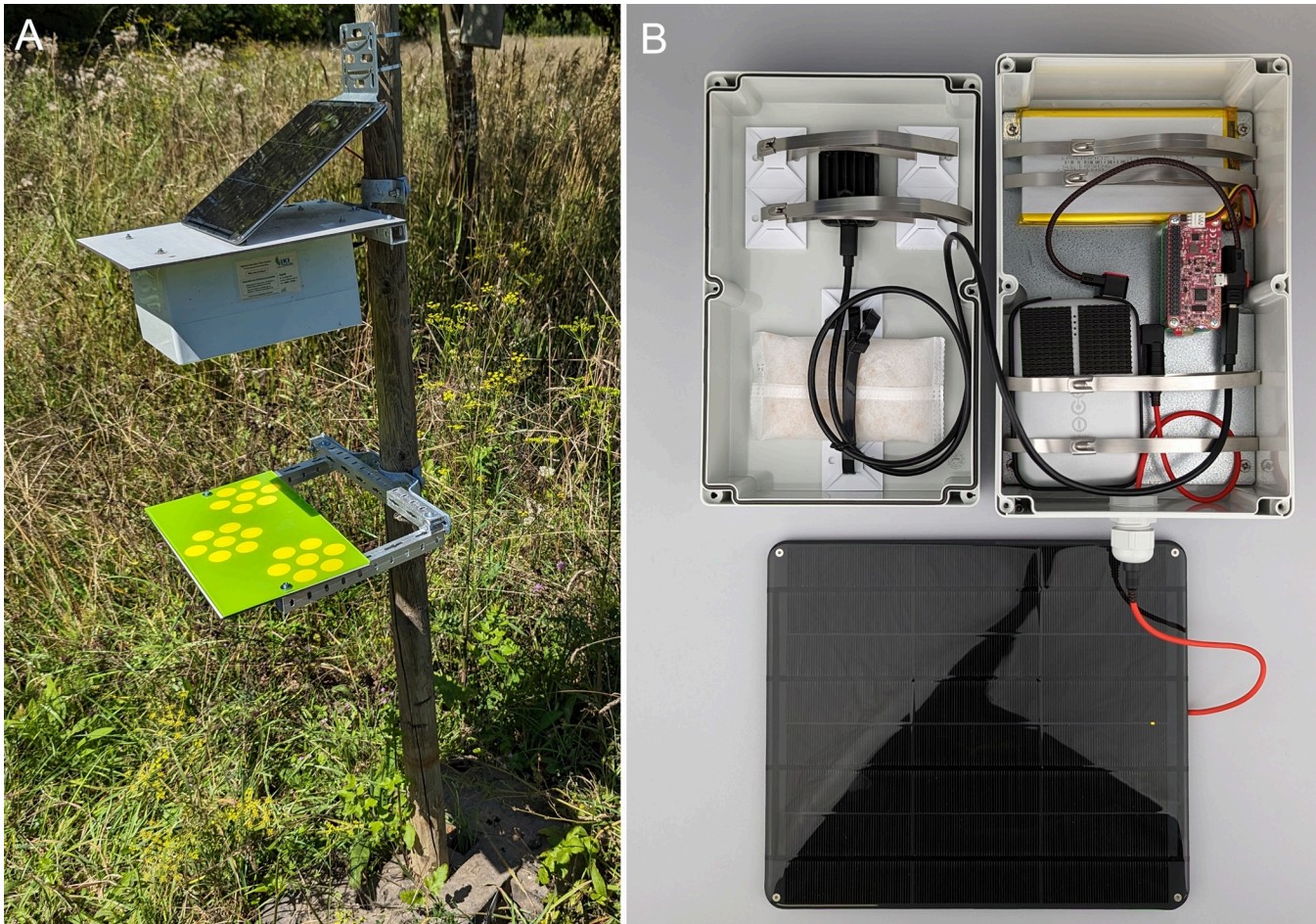

**Fig 1. Camera trap design.** (A) Field deployment of the camera trap and flower platform (35x20 cm) on wooden post. (B) Weatherproof camera trap enclosure with integrated hardware and connected solar panel.

the detection model and/or object tracker output, and for capturing high-resolution frames (1080p/4K/12MP) or videos (1080p/4K). All scripts use auto focus, auto exposure and auto white balance for the OAK-1 camera by default. Optionally, the auto focus range can be set by providing the minimum and maximum distance to which the auto focus should be restricted (in centimeters measured from the camera). To set the auto exposure region, the area (bounding box) of a detected insect can be used optionally. Both settings were only implemented recently and were therefore not used during collection of the data presented in the following.

## On-device insect detection

Together with the software, custom trained YOLOv5n [28], YOLOv6n [29], YOLOv7-tiny [30] and YOLOv8n [31] object detection models are provided, which can be run on the OAK-1 chip to enable real-time insect detection and tracking. The respective model weights, pre-trained on the MS COCO dataset [32], were fine-tuned on an image dataset, collected with a camera trap prototype in 2022 [33]. The dataset is composed of 1,335 images, of which 1,225 contain at least one annotation (110 background images). A total of 2,132 objects were annotated, including 664 wasps, 454 flies, 297 honey bees, 297 other arthropods, 233 shadows of insects and 187 hoverflies. For detection model training, all originally annotated classes were

**Table 1. Metrics of the YOLO insect detection models.**

| Model | Image size [pixel] | mAP$^{val}$ (@0.5 IoU) | Precision$^{val}$ | Recall$^{val}$ | Speed OAK-1 [fps] | Parameters (million) |
|---|---|---|---|---|---|---|
| YOLOv5n | 320 | 0.969 | 0.955 | 0.961 | 49 | 1.76 |
| YOLOv6n | 320 | 0.951 | 0.969 | 0.898 | 60 | 4.63 |
| YOLOv7-tiny | 320 | 0.957 | 0.947 | 0.942 | 52 | 6.01 |
| YOLOv8n | 320 | 0.944 | 0.922 | 0.899 | 39 | 3.01 |

All models were trained on a custom dataset with 1,335 images (1,069 in train split) to 300 epochs with batch size 32 and default hyperparameters. Metrics (mAP, Precision, Recall) are shown on the dataset validation split (133 images) for the original PyTorch (.pt) models before conversion to.blob format. Speed (fps) is shown for the converted.blob models running on the OAK-1 connected to the RPi Zero 2 W.

merged into one generic class ("insect"), the original 4K resolution (3840x2160 pixel) of the images was downscaled and stretched to 320x320 pixel, and the dataset was randomly split into train (1,069 images), validation (133 images) and test (133 images) subsets with a ratio of 0.8/0.1/0.1.

The provided YOLO models are general insect detectors and can also detect insects not included in the dataset and/or on other homogeneous backgrounds (e.g. variations of the artificial flower platform design). Due to the downscaled input image size of 320x320 pixel, the models can achieve a high on-device inference speed, while still keeping good precision and recall values (Table 1). The YOLOv5n model achieved the highest mAP (mean average precision) and recall on the dataset validation split and is used as default model by the camera trap software, with an IoU (intersection over union) and confidence threshold of 0.5 respectively. Google Colab notebooks are provided at GitHub [34] to reproduce the model training and validation, or train detection models on custom datasets.

## On-device processing pipeline

The main Python script for continuous automated insect monitoring uses a specific processing pipeline to run insect detection on downscaled LQ (low-quality) frames (320x320 pixel). This increases inference speed of the model and thereby accuracy of the object tracker that uses the coordinates from the model output as input for tracking insects. The object tracker is based on the Kalman Filter and Hungarian algorithm [35, 36], which can keep track of a moving object by comparing the bounding box coordinates of the current frame with the object's trajectory on previous frames. By assigning a unique ID to each insect landing on or flying above the flower platform, multiple counting of the same individual can be avoided as long as it is present in the frame. As the lack of visual features on LQ frames would significantly decrease classification accuracy, all tracked detections are synchronized with HQ (high-quality) frames (1920x1080 or 3840x2160 pixel) on-device in real time (Fig 2). The detected and tracked insects are then cropped from the synchronized HQ frames by utilizing the bounding box coordinates (Fig 3). The cropped detections are saved as individual.jpg images to the microSD card of the RPi together with relevant metadata including timestamp, confidence score and tracking ID with an image capture frequency of ~1 s. The image capture frequency can be adjusted optionally, without decreasing the overall pipeline and inference speed. Due to the synchronization of the tracker/model output with the HQ frames, the whole pipeline runs at ~12.5 fps for 1080p HQ frame resolution, or ~3.4 fps for 4K HQ frame resolution. Due to the higher possible inference speed of the detection model at 1080p HQ frame resolution, a higher tracking accuracy can be expected for fast-moving insects, potentially resulting in a more precise activity/abundance estimation. By using 4K HQ frame resolution, more details can be preserved in the cropped insect images, which can result in a higher classification

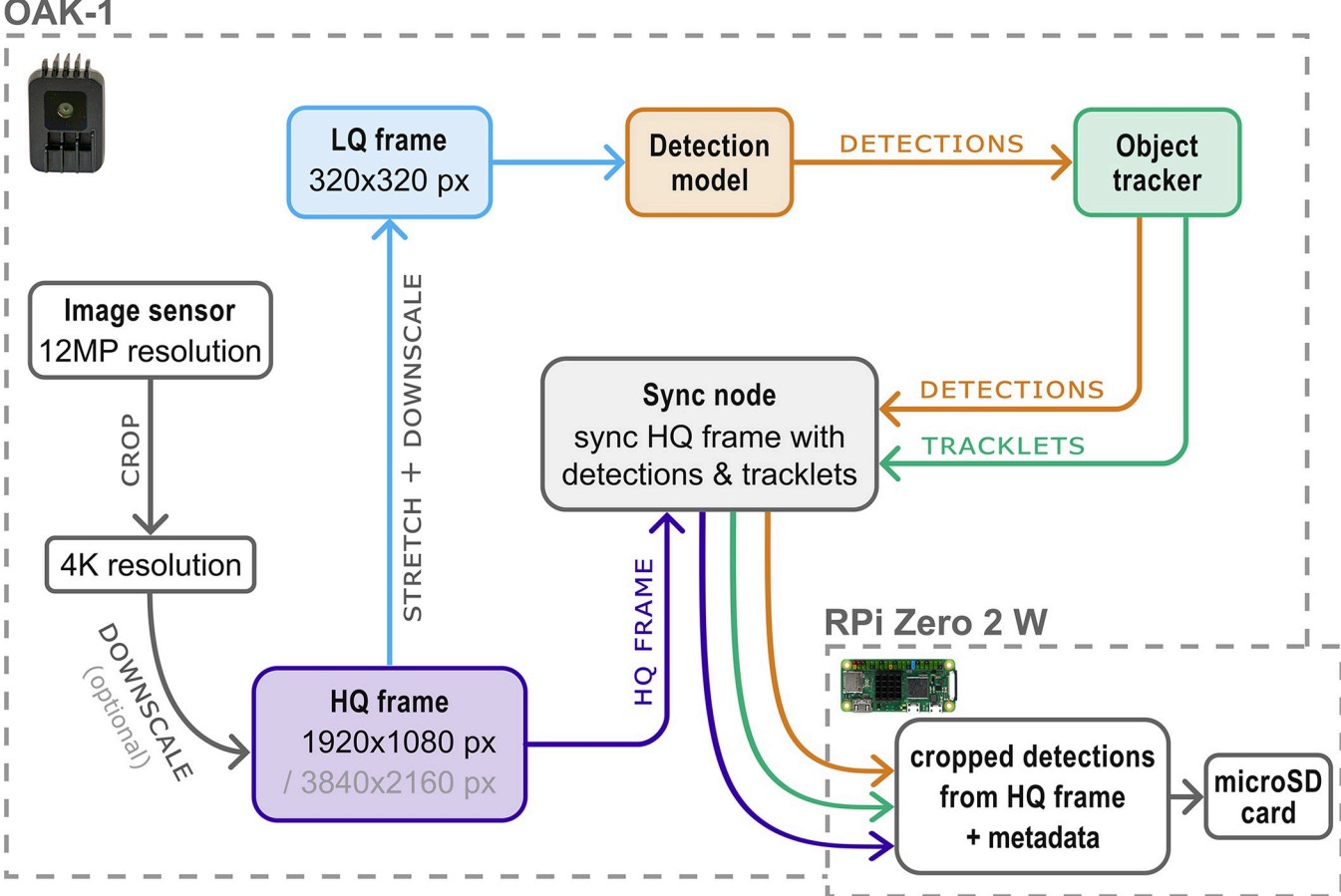

**Fig 2. Diagram of the processing pipeline.** HQ frames (1080p or 4K) are downscaled to LQ frames (320x320 pixel), while both HQ and LQ frames run in two parallel streams. The LQ frames are used as input for the insect detection model. The object tracker uses the coordinates from the model output to track detected insects and assign unique tracking IDs. Detected insects are cropped from synchronized HQ frames in real time and saved to the microSD card of the RPi together with relevant metadata (including timestamp, tracking ID, coordinates).

accuracy, especially for small species. This tradeoff should be carefully considered when selecting the HQ frame resolution.

The power consumption of the system was measured at room temperature with a USB power measuring device (JT-UM25C, SIMAC Electronics GmbH, Neukirchen-Vluyn, Germany) connected between the Voltaic 12,800mAh battery and the PiJuice Zero pHAT. The solar panel and PiJuice 12,000mAh battery were not connected to the system for this test, to avoid both components of influencing the measurement (e.g. additional charging of the PiJuice battery). The mean peak power consumption of the camera trap system is ~4.4 W, while constantly detecting and tracking five insects simultaneously, and saving the detections cropped from 1080p HQ frames to the RPi microSD card every second (Fig 4). With a combined battery capacity of ~91Wh, recordings can be run for ~20 hours, even if no sunlight is available for the solar panel to recharge the batteries. The PiJuice Zero pHAT is used for efficient power management. If the PiJuice battery charge level drops below a specified threshold, the RPi will immediately shut down before starting a recording. The respective duration of each recording interval is conditional on the current charge level, which can prevent gaps in monitoring data during periods with less sunlight. The scheduled recording times, charge level thresholds and recording durations can be easily adjusted for each use case.

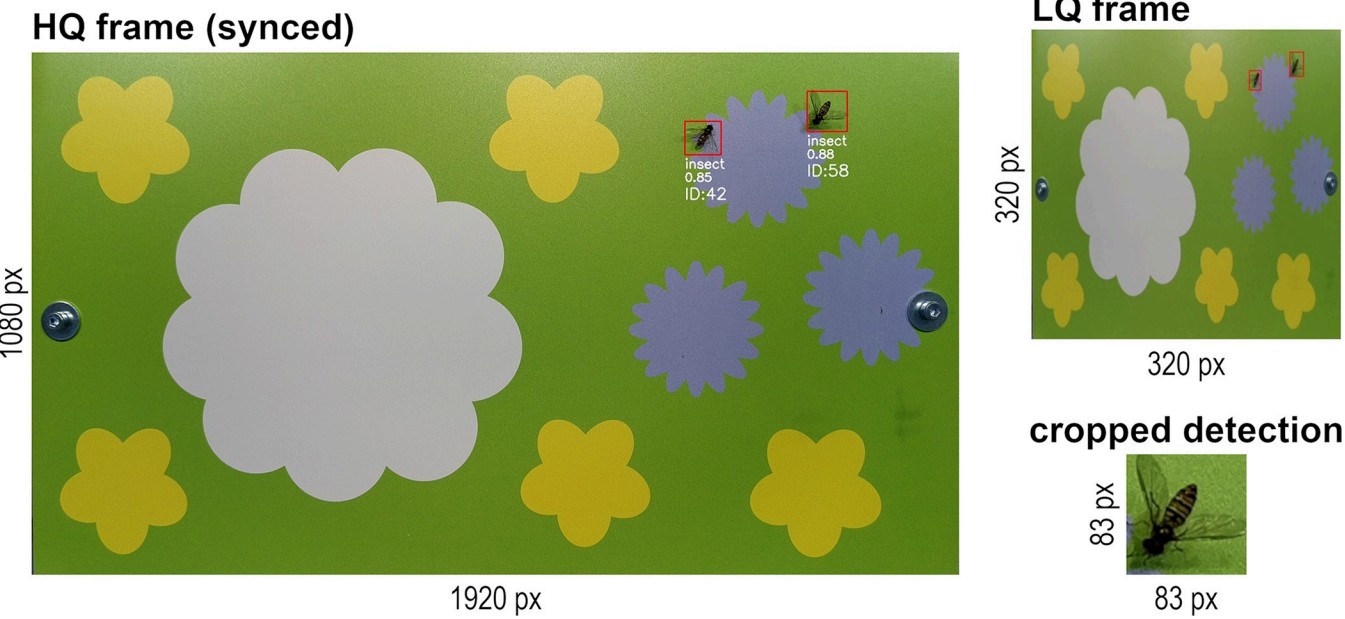

**Fig 3. LQ frame and synced HQ frame with cropped detection.** Downscaled LQ frames (320x320 pixel) are used as model input. Detected insects are cropped from synchronized HQ frames (1080p or 4K) on device.

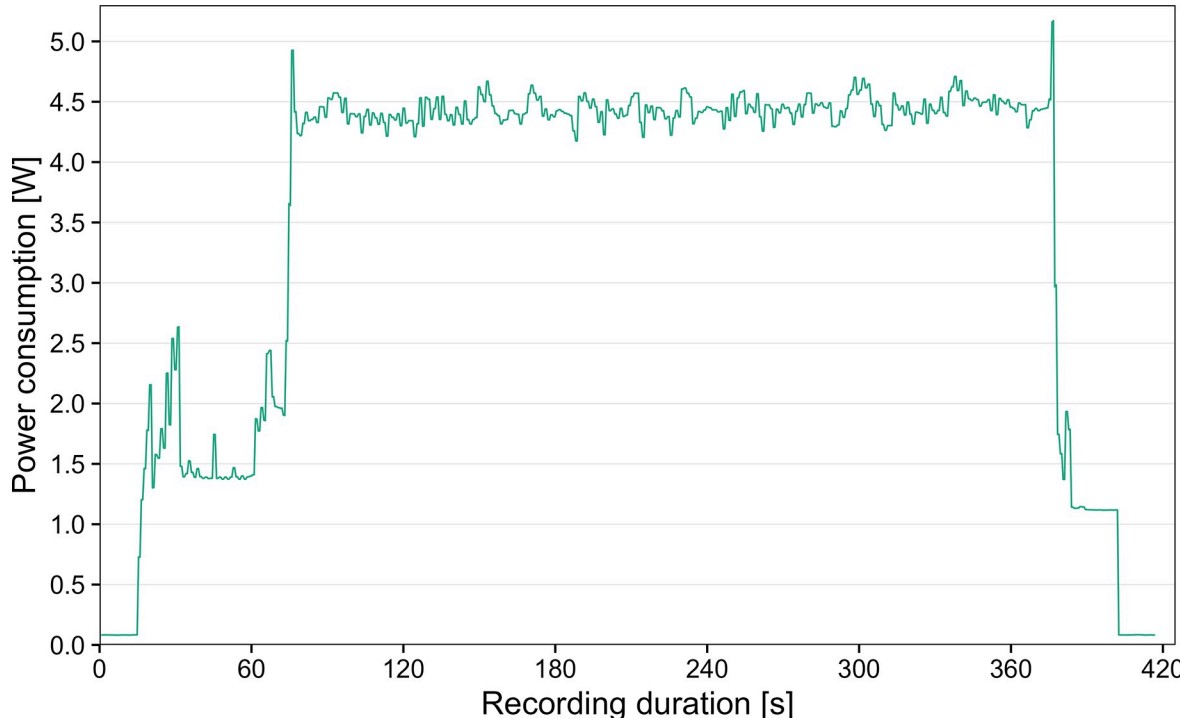

**Fig 4. Power consumption of the camera trap system.** The two power spikes at ~5W represent the start and end of the recording. Power consumption was measured during a 5 min recording interval while constantly detecting and tracking five insects, and saving the cropped detections together with metadata to the RPi microSD card every second.

The insect images are classified in a subsequent step, followed by post-processing of the metadata. This two-stage approach makes it possible to achieve high detection and tracking accuracy in real time on-device (OAK-1), combined with a high classification accuracy of the insect images cropped from HQ frames on a local computer.

## Insect classification

The data output from the automated monitoring pipeline includes insect images together with associated.csv files containing metadata for further analysis. Classification of the insect images is done in a subsequent step on a local computer, after the camera trap data has been collected from the microSD card. A modified YOLOv5 [28] script that supports several classification model architectures is used to classify all insect images captured with the camera trap and append the classification results (top-3 classes and probabilities) as new columns to the merged metadata. The script is available at GitHub [37] together with other modified YOLOv5 scripts, including classification model training and validation.

For training of the insect classification model, we used model weights, pre-trained on the ImageNet dataset [38]. The pre-trained weights were fine-tuned on a custom image dataset, mainly containing cropped detections captured by six camera traps [39]. The dataset is composed of 21,000 images, of which 18,597 images contain various arthropods (mainly insects). We sorted all images to 27 different classes, including four classes without insects ("none_*"). These additional classes are used to filter out images of background, shadows, dirt (e.g. leaves and bird droppings) and birds that can be captured by the camera trap. Most of the images were detections cropped from 1080p HQ frames, which were automatically collected with six different camera traps deployed in the field in 2022 and 2023. Additional images of some classes with insufficient occurrences in the field data (including "ant", "bee_bombus", "beetle_cocci", "bug", "bug_grapho", "hfly_eristal", "hfly_myathr", "hfly_syrphus") were automatically captured (cropped from 1080p HQ frames) with a lab setup of the camera trap hardware to increase accuracy and generalization capability of the model. Detailed descriptions of all classes, which were not only chosen by taxonomical but also visual distinctions, can be found in S1 Table and example images are shown in Fig 5.

For model training, the dataset was randomly split into train (14,686 images), validation (4,189 images) and test (2,125 images) subsets with a ratio of 0.7/0.2/0.1. We compared the metrics of three different model architectures (YOLOv5-cls [28], ResNet-50 [40], EfficientNet-B0 [41]) supported by YOLOv5 classification model training and different hyperparameter settings to find the combination with the highest accuracy on our dataset (S2 Table). We selected the EfficientNet-B0 model trained to 20 epochs with batch size 64 and images scaled to 128x128 pixel, as it achieved an overall high accuracy on the dataset validation and test split, while generalizing better to a real-world dataset compared to the same model trained to only ten epochs. With this model, high top-1 accuracies on the test dataset for all insect classes could be achieved (Fig 6, Table 2). Only three classes ("beetle", "bug", "other") did not reach a top-1 accuracy > 0.9, probably because of a high visual intraclass heterogeneity and not enough images in the dataset.

The insect classification model was exported to ONNX format, which enables faster inference speed on standard CPUs (~10–20 ms per image), and is available at GitHub [34]. To reproduce the model training and validation, or train classification models on custom datasets, Google Colab notebooks are provided in the same repository [34].

## Metadata post-processing

By running the insect classification, all available metadata files from the camera trap output are merged and the top-3 classification results for each insect image are appended to new

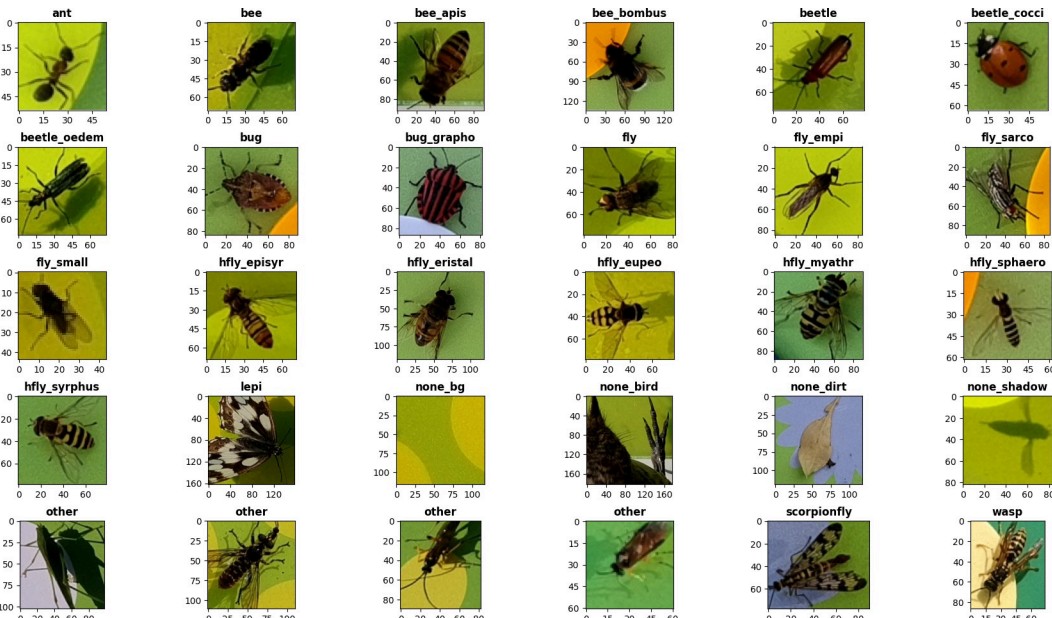

**Fig 5. Example images of the 27 classes from the dataset for classification model training.** All images were captured automatically by the camera trap and are shown unedited, but resized to the same dimension. Original pixel values are plotted on the x- and y-axis for each image. Four example images are shown for the class "other", to account for its high intraclass heterogeneity.

columns. In most cases, the merged metadata still contains multiple rows for each insect with a unique tracking ID, with the number of rows depending on the image capture frequency (default: ~1 s) and on the duration the insect was present in the frame. With multiple captured images for each individual insect, classification of the images can result in different top-1 classes for the same tracking ID. The Python script for post-processing and analysis [34] is used to generate the final metadata file that can then be used for further data analysis. The weighted mean classification probability is calculated to determine the top-1 class with the overall highest probability for each tracking ID, by calculating the mean probability of each top-1 class per tracking ID and multiplying it with the proportion of images classified to the respective top-1 class to the total number of images per tracking ID:

$$weighted\ mean\ probability_{top1\ class} = mean\Big(probability_{top1\ class}\Big) \times \frac{images_{top1\ class}}{images_{tracking\ ID}}$$

Only the top-1 class with the highest weighted mean classification probability per tracking ID is kept in the final metadata file, in which each row corresponds to an individual tracked insect.

To make the activity/abundance estimations more reliable, all individual insects (= unique tracking IDs) with less than three or more than 1,800 images are excluded by default. These values can be adjusted optionally, e.g. if a different image capture frequency is used and/or to make the final data more robust to inaccurate tracking of fast-moving insects. This can result in "jumping" tracking IDs, as the limited inference speed of the detection model is providing the object tracker with bounding box coordinates in a frequency that is too low to keep track of the fast-moving insect (S2B Fig). In this case, the track of the insect is lost and a new tracking ID is assigned to the same individual. With the default image capture frequency of ~1 s, the insect had to be tracked for at least slightly over two seconds to be kept in the final dataset.

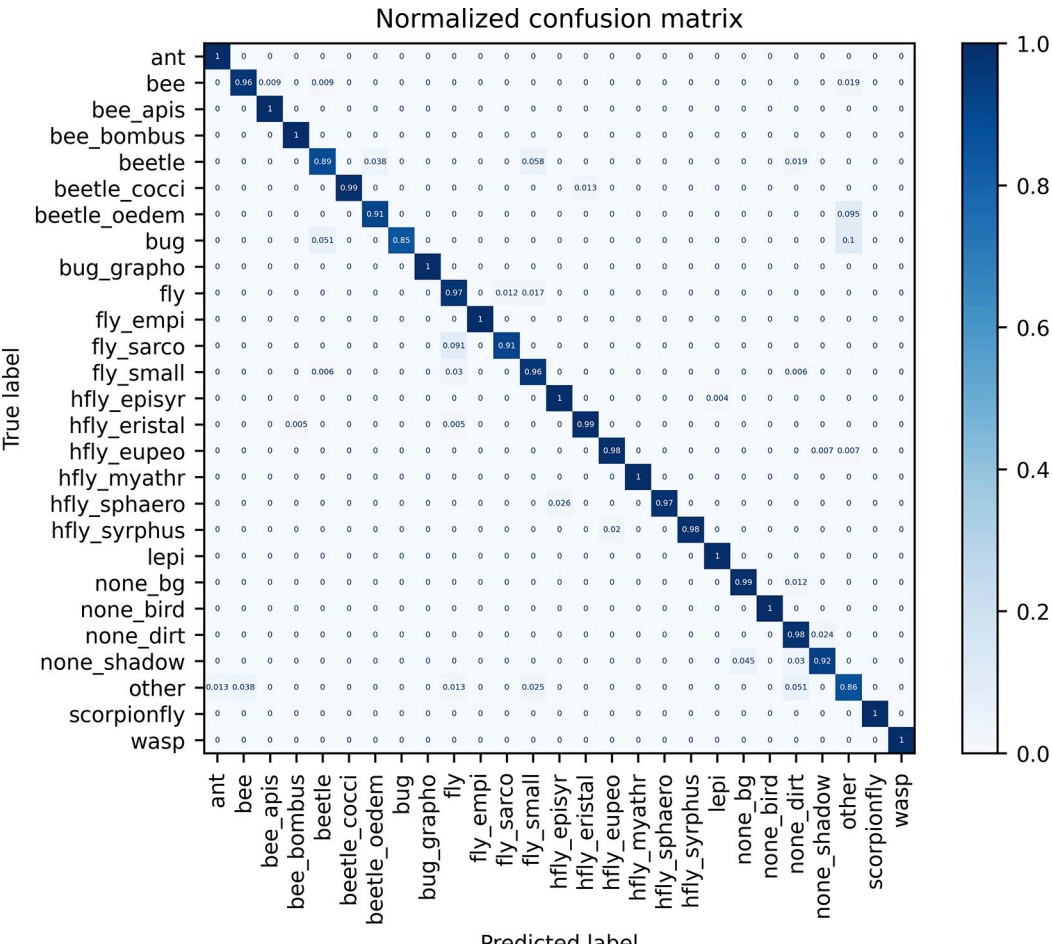

**Fig 6. Normalized confusion matrix for the EfficientNet-B0 insect classification model, validated on the dataset test split.** The cell values show the proportion of images that were classified to a predicted class (x-axis) to the total number of images per true class (y-axis). The model was trained on a custom dataset with 21,000 images (14,686 in train split). Metrics are shown on the dataset test split (2,125 images) for the converted model in ONNX format.

The default upper limit of 1,800 images per tracking ID removes all IDs that were tracked for more than ~30–40 min at the default image capture frequency. This maximum duration depends on the number of simultaneous detections, which can slightly decrease the capture frequency over time. From our experience, objects tracked for > 30 min are most often non-insect detections, e.g. leaves fallen on the flower platform.

For estimation of the respective insect size, the absolute bounding box sizes in millimeters are calculated by supplying the true frame width and height (e.g. flower platform dimensions).

## Insect tracking evaluation

The accuracy of the object tracker was tested in a lab experiment. The camera trap hardware was installed 40 cm above the big flower platform (50x28 cm), so that the platform was filling out the whole camera frame, and both were placed in the middle of a 200x180x155 cm insect cage. The script for continuous automated insect monitoring was run for 15 min for each recording interval. We used the slower synchronization with 4K HQ frame resolution (~3.4 fps) to test the more difficult use case for the object tracker, compared to the faster 1080p

**Table 2. Metrics of the EfficientNet-B0 insect classification model, validated on the dataset test split.**

| Class | Images<sup>test</sup> | Top-1 accuracy<sup>test</sup> | Precision<sup>test</sup> | Recall<sup>test</sup> | F1 score<sup>test</sup> |
|---|---|---|---|---|---|
| all | 2125 | 0.972 | 0.971 | 0.967 | 0.969 |
| ant | 111 | 1.0 | 0.991 | 1.0 | 0.996 |
| bee | 107 | 0.963 | 0.972 | 0.963 | 0.967 |
| bee_apis | 31 | 1.0 | 0.969 | 1.0 | 0.984 |
| bee_bombus | 127 | 1.0 | 0.992 | 1.0 | 0.996 |
| beetle | 52 | 0.885 | 0.92 | 0.885 | 0.902 |
| beetle_cocci | 78 | 0.987 | 1.0 | 0.987 | 0.994 |
| beetle_oedem | 21 | 0.905 | 0.905 | 0.905 | 0.905 |
| bug | 39 | 0.846 | 1.0 | 0.846 | 0.917 |
| bug_grapho | 19 | 1.0 | 1.0 | 1.0 | 1.0 |
| fly | 173 | 0.971 | 0.944 | 0.971 | 0.957 |
| fly_empi | 19 | 1.0 | 1.0 | 1.0 | 1.0 |
| fly_sarco | 33 | 0.909 | 0.938 | 0.909 | 0.923 |
| fly_small | 167 | 0.958 | 0.952 | 0.958 | 0.955 |
| hfly_episyr | 253 | 0.996 | 0.996 | 0.996 | 0.996 |
| hfly_eristal | 197 | 0.99 | 0.995 | 0.99 | 0.992 |
| hfly_eupeo | 137 | 0.985 | 0.993 | 0.985 | 0.989 |
| hfly_myathr | 60 | 1.0 | 1.0 | 1.0 | 1.0 |
| hfly_sphaero | 39 | 0.974 | 1.0 | 0.974 | 0.987 |
| hfly_syrphus | 50 | 0.98 | 1.0 | 0.98 | 0.99 |
| lepi | 24 | 1.0 | 0.96 | 1.0 | 0.98 |
| none_bg | 86 | 0.988 | 0.966 | 0.988 | 0.977 |
| none_bird | 8 | 1.0 | 1.0 | 1.0 | 1.0 |
| none_dirt | 85 | 0.976 | 0.902 | 0.976 | 0.938 |
| none_shadow | 66 | 0.924 | 0.953 | 0.924 | 0.938 |
| other | 79 | 0.861 | 0.883 | 0.861 | 0.872 |
| scorpionfly | 12 | 1.0 | 1.0 | 1.0 | 1.0 |
| wasp | 52 | 1.0 | 1.0 | 1.0 | 1.0 |

The model was trained on a custom dataset with 21,000 images (14,686 in train split). Metrics are shown on the dataset test split (2,125 images) for the converted model in ONNX format.

resolution (~12.5 fps). 15 *Episyrphus balteatus* hoverflies (reared under lab conditions) were released inside the cage and recorded for 15 min for each of the ten replications. The flower platform was simultaneously filmed with a smartphone camera (1080p, 30 fps) during each recording interval. The videos were played back afterwards with reduced speed (25–50%) and all frame/platform visits of the hoverflies were counted manually. The captured metadata was post-processed with ten different settings for the minimum number of images required per tracking ID to be kept in the final data output. The true platform visits from the video count were then compared with the number of unique tracking IDs for each setting.

## Insect classification validation

While the metrics of the custom trained EfficientNet-B0 classification model show a high accuracy on the dataset test split, the underlying dataset was curated and insect images were selected in such a way to include images where the respective class was clearly identifiable and also some images with more difficult cases, e.g. with only a part of the insect visible. For a more realistic measure of the classification accuracy and an estimation of the generalization

capability of the model, we compiled a dataset with images captured during field deployment of five camera traps between August 11 and September 18, 2023. All images were automatically cropped detections from 1080p HQ frames. No images from this recording period were included in the dataset for classification model training. All captured 93,215 images from camera trap 1 were classified and subsequently manually verified, to ensure that all images were sorted to the correct class. As some classes were only present with a small number of images, more images from camera traps 2–5 were added to achieve a better class balance, resulting in a total dataset size of 97,671 images. All images from each class were added to the dataset, including false negative classifications (image of target class classified to wrong class) and false positive classifications (image of other class classified wrongly to target class). No images were removed to reflect a real-world use case, which means that many nearly identical images are present in the dataset (e.g. insects moving only slightly or none-insect detections such as leaves) and the overall class balance is biased towards classes with more captured images. For two classes ("bug_grapho", "fly_empi"), no images were captured during the selected recording period. To run the model validation, a dummy image for each of these classes was added to the dataset and thus results for both classes must be ignored.

## Field deployment

Starting in mid-May 2023, five camera traps were deployed at five different sites in southwestern Germany. All sites were located within a range of 50 km and separated by at least 5 km, with an elevation ranging from 90–170 m.a.s.l. As study sites, extensively managed orchard meadows were chosen, where an insect monitoring with focus on hoverflies using traditional methods (Malaise traps, yellow pan traps) was also being conducted. While the area, age and tree density differed between the sites, all meadows were cut at least once per year.

All camera traps were positioned in full sunlight with the solar panel and flower platform facing to the south. Scheduled recording intervals with a respective duration of 40 min (if the current battery charge level was > 70%) were run four to seven times per day, resulting in a recording duration of 160 to 280 min for each camera trap per day. Recordings started at 6, 7, 8, 10, 11, 14, 16, 18 and 19 o'clock, with the start time differing per month and in some cases also per camera trap. Due to differences in the date of first deployment (camera trap 3, camera trap 5) and initial hardware problems (camera trap 1), the total recording times differed between all camera traps (Table 3). To increase power efficiency and avoid potential recording gaps during times of low sunlight and decreased charging of the batteries, the respective duration of each recording interval is conditional on the current battery charge level. This additionally influenced the total recording time per camera trap, depending on the amount of sunlight available to charge the batteries. During field deployment, the artificial flower platform was changed at least once for each camera trap (except camera trap 5) to test different sizes, materials and designs (flower shapes and colors). Therefore, the results presented below should be interpreted with a certain degree of caution in relation to the insect monitoring data, and should be considered primarily as proof of concept for the camera trap system.

**Table 3. Recording times of the five deployed camera traps and site details of the orchard meadows.**

| Camera trap ID | Total recording time [h] | Site ID | Coordinates | Area [ha] |
|---|---|---|---|---|
| Camtrap 1 | 362 | Gross-Rohrheim | 49.718741, 8.509769 | 1.4 |
| Camtrap 2 | 456 | Boehl-Iggelheim | 49.379616, 8.330233 | 0.7 |
| Camtrap 3 | 349 | Malsch | 49.236468, 8.670059 | 0.9 |
| Camtrap 4 | 421 | Bensheim | 49.703971, 8.590160 | 1.7 |
| Camtrap 5 | 330 | Dossenheim | 49.446449, 8.649950 | 5.4 |

### Data visualization

Data analysis and creation of plots presented in the following was conducted with R version 4.3.1 [42] and the R packages tidyverse version 2.0.0 [43], patchwork version 1.1.3 [44] and viridis version 0.6.4 [45]. All required R scripts and associated data to reproduce the plots are available at Zenodo [46].

## Results

### Insect tracking evaluation

With 10 replications of 15 min recording intervals, the object tracker accuracy was tested together with cropping detected insects from synchronized 4K HQ frames (~3.4 fps) every second. High activity with many frame/platform visits of the hoverflies during the 15 min recording interval resulted in a decreased tracking accuracy, mainly due to hoverflies flying fast and erratic and/or coming close to each other (S2 Fig). This behavior can lead to "jumping" tracking IDs and consequently multiple counting of the same individual, which is reflected in the number of unique tracking IDs if none are removed during post-processing with the setting "-min_tracks 1" (Fig 7). By excluding tracking IDs with less than a specified number of images during post-processing, the activity/abundance estimation can get more robust. Even with a decreased pipeline speed of ~3.4 fps during synchronization with 4K HQ frames, an exclusion of tracking IDs with less than six images ("-min_tracks 6") led to a relatively precise estimation of hoverfly activity/abundance in the data from the lab experiment (Fig 7).

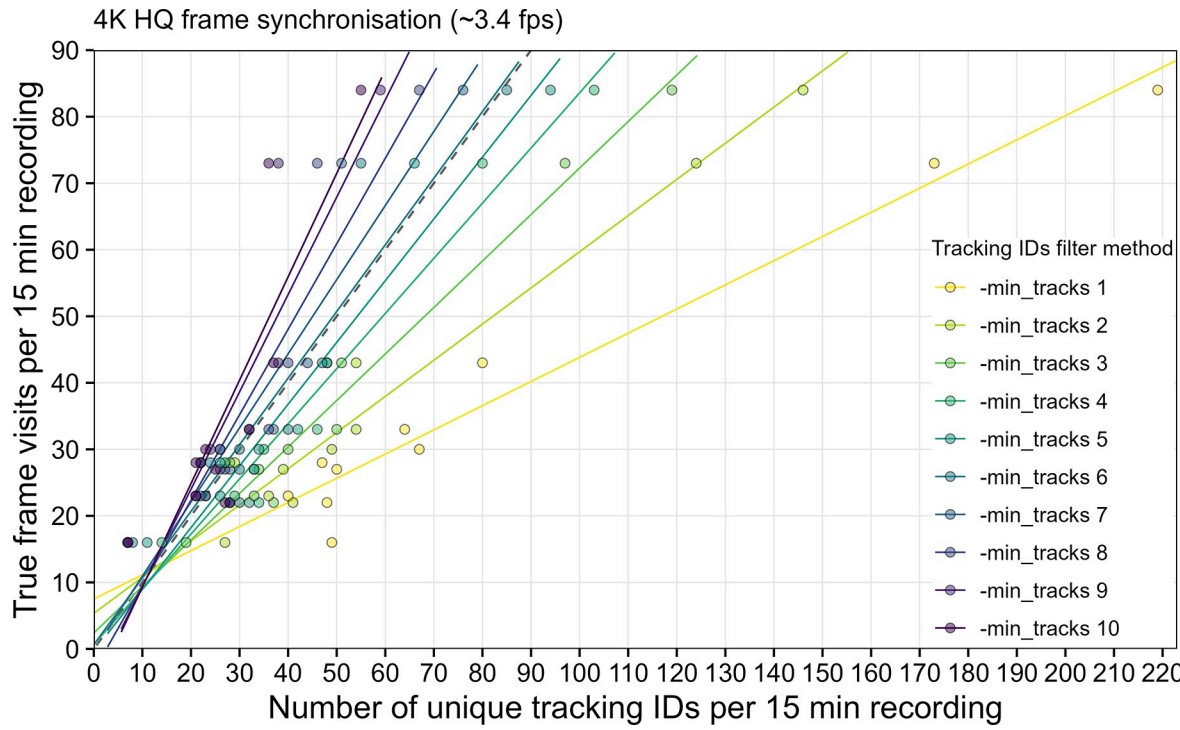

**Fig 7. Evaluation of the insect tracking accuracy in a lab experiment.** Data from 10 replications of 15 min recording intervals with 15 *E. balteatus* hoverflies placed in a cage with the camera trap and flower platform is shown. Linear regression lines illustrate the effect of 10 different tracking IDs filter methods for post-processing of the captured metadata of each recording interval. With "-min_tracks 1" no tracking IDs are excluded. Dashed line indicates optimal result. True frame visits were manually counted from video recordings of the flower platform.

## Insect classification validation

To estimate the generalization capability of the custom trained EfficientNet-B0 insect classification model, a dataset with images captured between August 11 and September 18, 2023 was compiled. For some classes ("beetle", "bug", "other", "wasp") classification accuracy was very low (Fig 8, S3 Table). For the classes "beetle_cocci" and "beetle_oedem", some of the images were classified as "beetle", which is the correct taxonomic order and could therefore be analyzed and interpreted without drawing overly wrong conclusions. In cases of uncertainty and

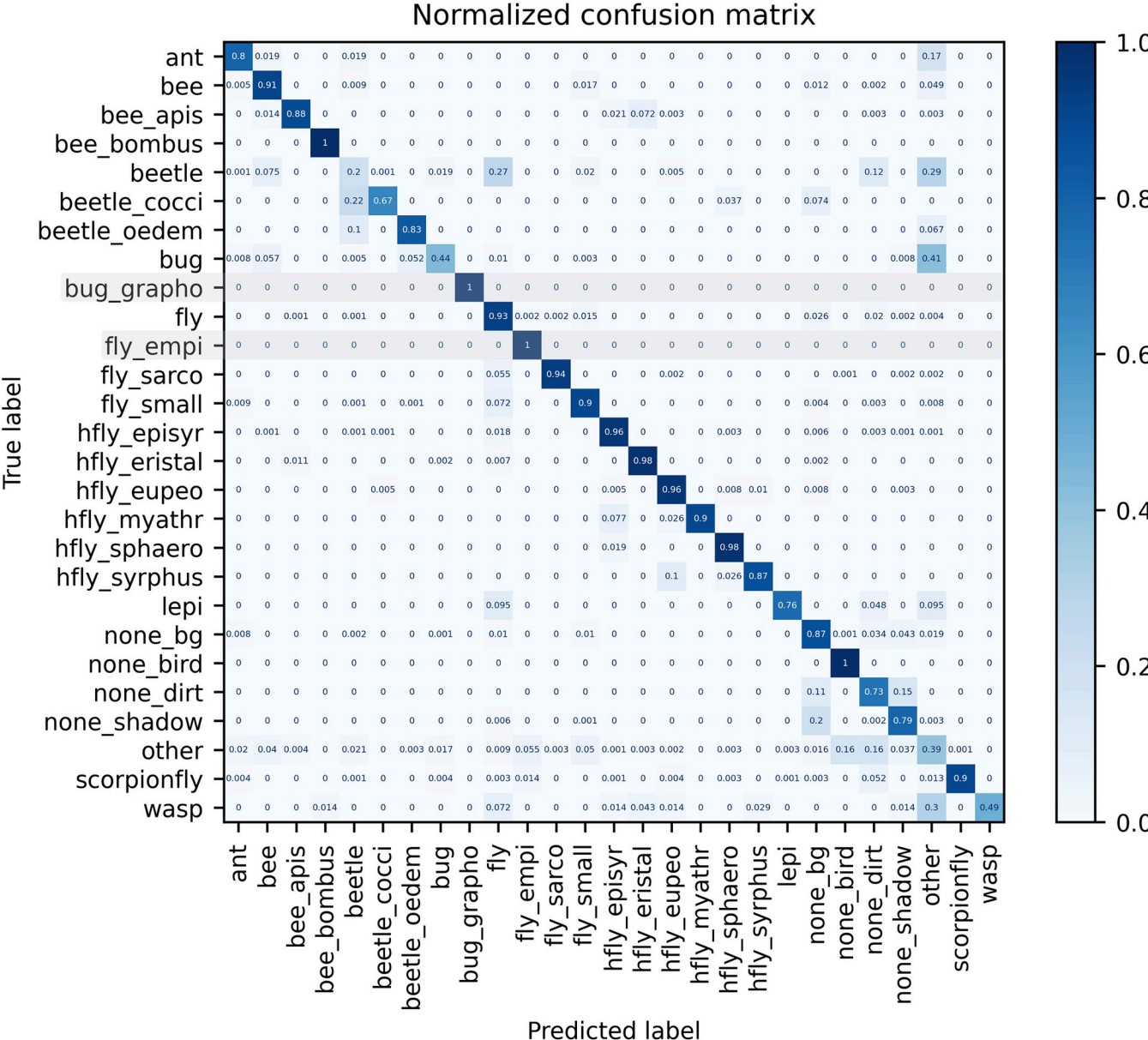

**Fig 8. Normalized confusion matrix for the EfficientNet-B0 insect classification model, validated on a real-world image dataset.** The cell values show the proportion of images that were classified to a predicted class (x-axis) to the total number of images per true class (y-axis). Metrics are shown on a real-world dataset (97,671 images) for the converted model in ONNX format. All images were classified and subsequently verified and sorted to the correct class in the case of a wrong classification by the model. A dummy image was added for each of the classes "bug_grapho" and "fly_empi", as no images of both were captured. Results for both classes must be ignored.

wrong model predictions, images seem to have a tendency to be classified to the class "other", which has a high intraclass heterogeneity in the training dataset. For our focus group of hover-flies ("hfly_*"), a high classification accuracy was achieved for all classes of species (-groups). Some images of *Syrphus* sp. hoverflies ("hfly_syrphus") were wrongly predicted as *Eupeodes* sp. ("hfly_eupeo"), which could be explained by the high visual similarity of both genera. Over-all, classes with a high visual intraclass heterogeneity, such as "beetle", "bug" and "other" could not be classified with a sufficient accuracy in the real-world dataset. Classes with less intraclass heterogeneity achieved a high classification accuracy in most cases.

## Field deployment

Five camera traps were deployed at five different sites in southwestern Germany, starting in mid-May 2023. The capability of the camera traps to withstand high temperatures and humid-ity was tested, as well as the performance of the solar panel and two connected batteries used as power supply for the system. The captured insect images were classified with the custom trained EfficientNet-B0 classification model, and metadata was post-processed with the default settings to exclude all tracking IDs with less than three or more than 1,800 images.

## Weather resistance

The maximum air temperature measured at the nearest weather station during deployment of the camera traps was 37.5°C in July. The maximum temperature measured during the camera trap recordings was 81°C for the OAK-1 CPU and 66°C for the RPi CPU (Fig 9). Both mea-surements still lie in the safe operating temperature range for these devices and a normal func-tionality of the camera trap can be expected if the air temperature does not exceed ~38°C.

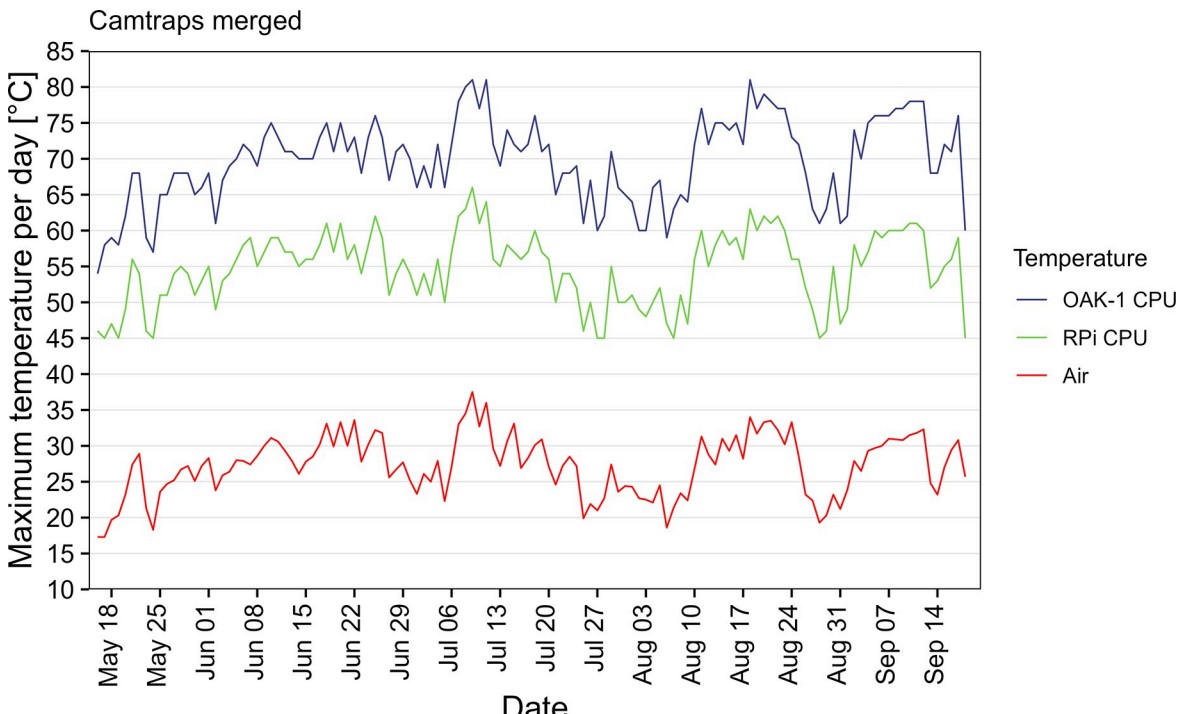

**Fig 9. Maximum air and OAK-1/RPi CPU temperatures per day.** Weather data was taken from the nearest weather station (source: Deutscher Wetterdienst).

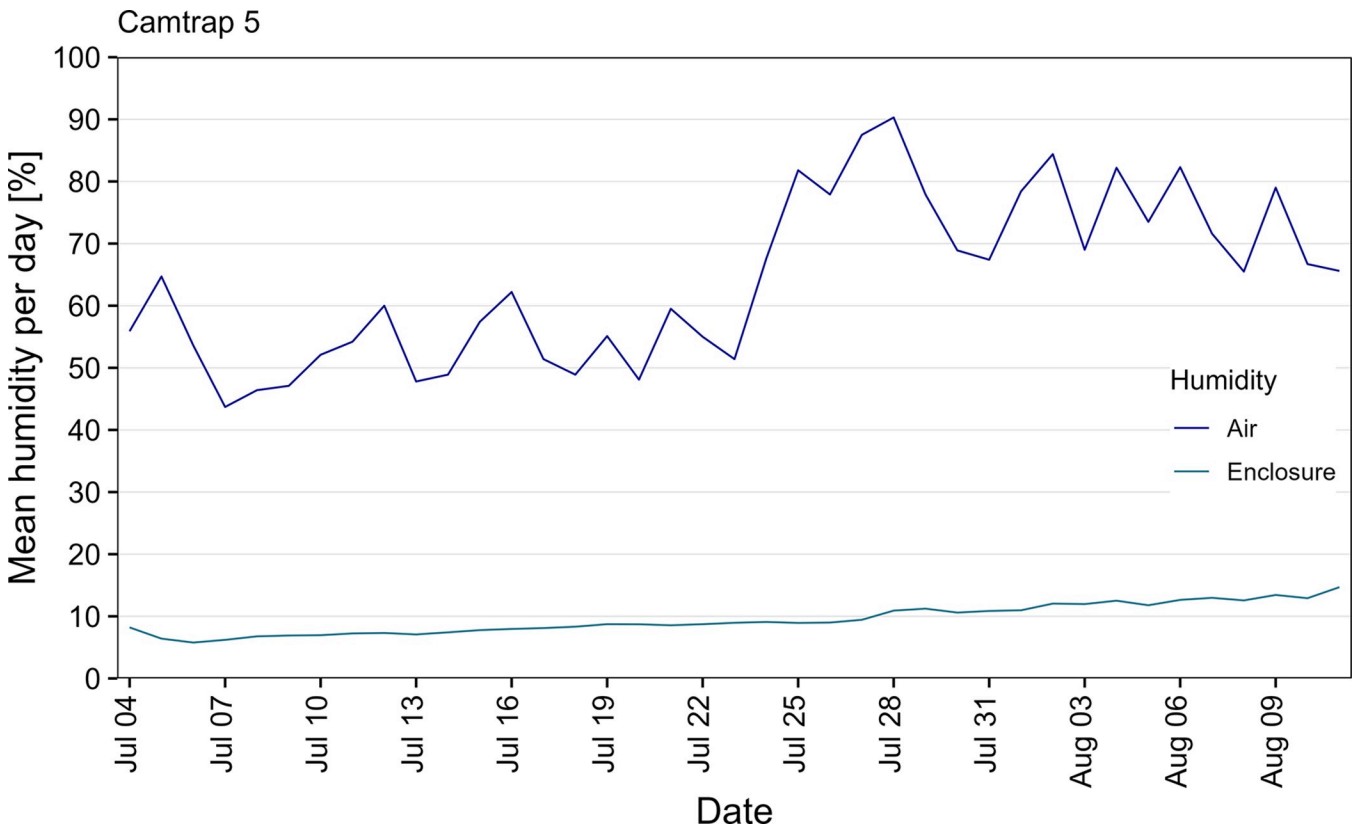

**Fig 10. Mean humidity per day measured inside the enclosure and at nearest weather station.** Weather data was taken from the nearest weather station (source: Deutscher Wetterdienst).

For camera trap 5, a humidity/temperature USB logger was placed inside the enclosure from July to mid-August. Even during days with a high mean air humidity, measured at the nearest weather station, the mean humidity inside the enclosure only increased slightly to a maximum of ~15% (Fig 10). For all five camera traps, no buildup of moisture during the recording time until mid-September could be noticed, even without exchanging the originally installed 50g Silica gel pack.

Two rechargeable batteries connected to a 9W solar panel are used as power supply for the camera trap system. During sunny days with a sunshine duration of ~8–12 h per day, the charge level of the PiJuice battery stayed relatively constant at > 80% for all five camera traps (Fig 11). A drop in the battery charge level can be noticed during end of July and end of August, when the sunshine duration per day decreased to ~0–4 h for several days. Since the duration of each recording interval is conditional on the current charge level, decreased recording durations preserved battery charge even when less sunlight was available to recharge the batteries. A fast recovery to charge levels > 80% restored the normal recording durations within a few days, after the sunshine duration increased again.

## Insect monitoring data

During a recording time of ~1919 h of all five camera traps combined, a total of ~2.34 million images with 49,583 unique tracking IDs were captured between mid-May and mid-September 2023. After post-processing of the metadata with default settings, 23,900 tracking IDs with less than three images (= tracked less than ~2 s) and 85 tracking IDs with more than 1,800 images

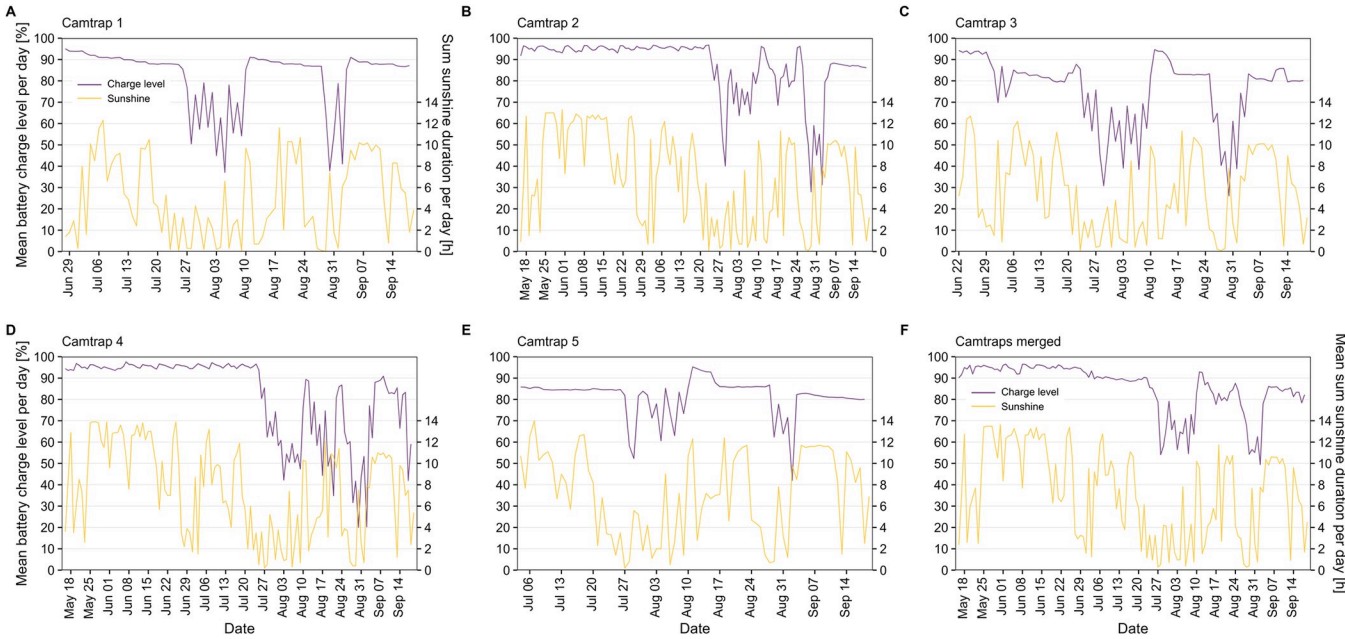

**Fig 11. Mean PiJuice battery charge level and sum of the sunshine duration per day.** The PiJuice battery was charged by a second battery, connected to a 9W solar panel. Weather data was taken from the nearest weather station (source: Deutscher Wetterdienst).

(= tracked longer than ~30–40 min) were removed from the final dataset, resulting in ~2.03 million images with 25,598 unique tracking IDs.

Out of the 25,598 unique tracking IDs, 8,677 were classified to one of the non-insect classes ("none_bg", "none_bird", "none_dirt", "none_shadow") (S3 Fig). Flies ("fly_small", "fly") were the most frequently captured insects, followed by wild bees excluding *Bombus* sp. ("bee") and other arthropods ("other") (Fig 12F). For hoverflies, 1,090 unique tracking IDs of *E. balteatus* ("hfly_episyr"), followed by 672 unique tracking IDs of *Eupeodes corollae* or *Scaeva pyrastri* ("hfly_eupeo") and 220 unique tracking IDs of the other four hoverfly classes were recorded by all camera traps. Differences in the composition of the captured insects between the camera traps can be observed, which could have been influenced by site conditions and/or by different flower platforms used for testing purposes during field deployment.

When an insect leaves the frame/platform and re-enters it again, a new tracking ID is assigned with the risk to count the same individual multiple times, which could influence the activity/abundance estimations. To assess this risk, the minimum time difference to the previous five tracking IDs that were classified as the same insect class was calculated for each unique tracking ID. A total of 3,150 tracking IDs showed a time difference of less than five seconds to the previous tracking ID that was classified as the same class (Fig 13). This equals ~18.6% of the 16,921 total tracking IDs classified as insects in our present dataset, of which some might have been the same individuals re-entering the frame.

Data from the images classified as one of the hoverfly classes is presented in more detail in the following plots, as an example for a functional group of special interest. Overall, the number of captured hoverfly tracking IDs varied throughout May and June, but also the total recording time per day was comparatively low during this period (Fig 14F). A peak in late June/early July is followed by steady numbers throughout July. Less hoverflies were recorded in August, though the recording time per day was also decreased during the first half of the month, due to lower battery charge levels caused by less available sunshine. Significantly lower

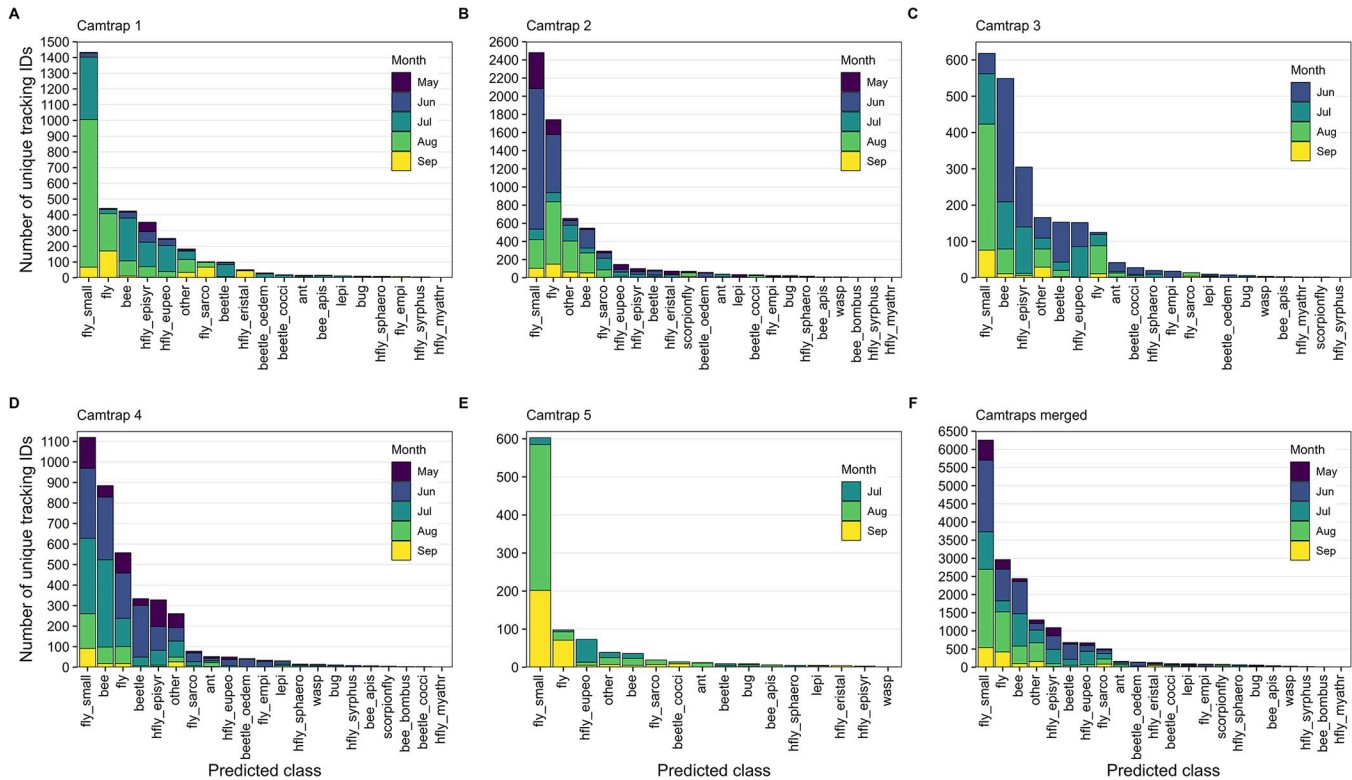

**Fig 12. Total number of unique tracking IDs for each predicted insect class.** (A–E) Data from camera traps 1–5. (F) Merged data from all camera traps. Data of non-insect classes not shown. All tracking IDs with less than three or more than 1,800 images were removed.

numbers of hoverflies were captured by camera trap 5 compared to the other four camera traps (Fig 14E). While the maximum recording time per day was normally 280 min for each camera trap, an overlap of changed recording start times resulted in 320 min total recording time per day for camera trap 3 in June 28 (Fig 14C).

With differences in the total recording time per day between days and camera traps, the activity, calculated as the number of unique tracking IDs per hour of active recording, can be a more adequate estimation of hoverfly activity/abundance. While the range of hoverfly activity was mostly similar between all camera traps, a peak in activity at the first day of deployment (May 16) can be observed for camera trap 1 (Fig 15A). In total, 41 unique tracking IDs of hoverflies were captured during a single 40 min recording interval at that day, which extrapolates to 62 tracking IDs per hour. This initially high activity could have been caused by a smaller group of hoverflies visiting the platform multiple times due to a higher attractivity induced by novelty. The highest hoverfly activity for a prolonged period was recorded by camera trap 3 in early July (Fig 15C). A decrease in hoverfly activity for all camera traps in late July/early August could have been caused by more rainfall during this time (Fig 15F).

The start of the recording intervals was scheduled at 6, 7, 8, 10, 11, 14, 16, 18 and 19 o'clock, with the respective start times differing per month and in some cases also per camera trap. This resulted in differences in the total recording time per hour, with most recordings available from 8, 10, 16 and 18 o'clock, and only few recordings available from 14 o'clock (S4 Fig). Overall, the highest hoverfly activity was measured at 8 and 10 o'clock, with approximately two unique hoverfly tracking IDs captured per hour on average (Fig 16F). About one hoverfly was captured per hour at 7 and 16 o'clock. A similarity in high activity during hours before noon compared to the other times of day can be seen also for bees (S5 Fig) and flies (S6 Fig).

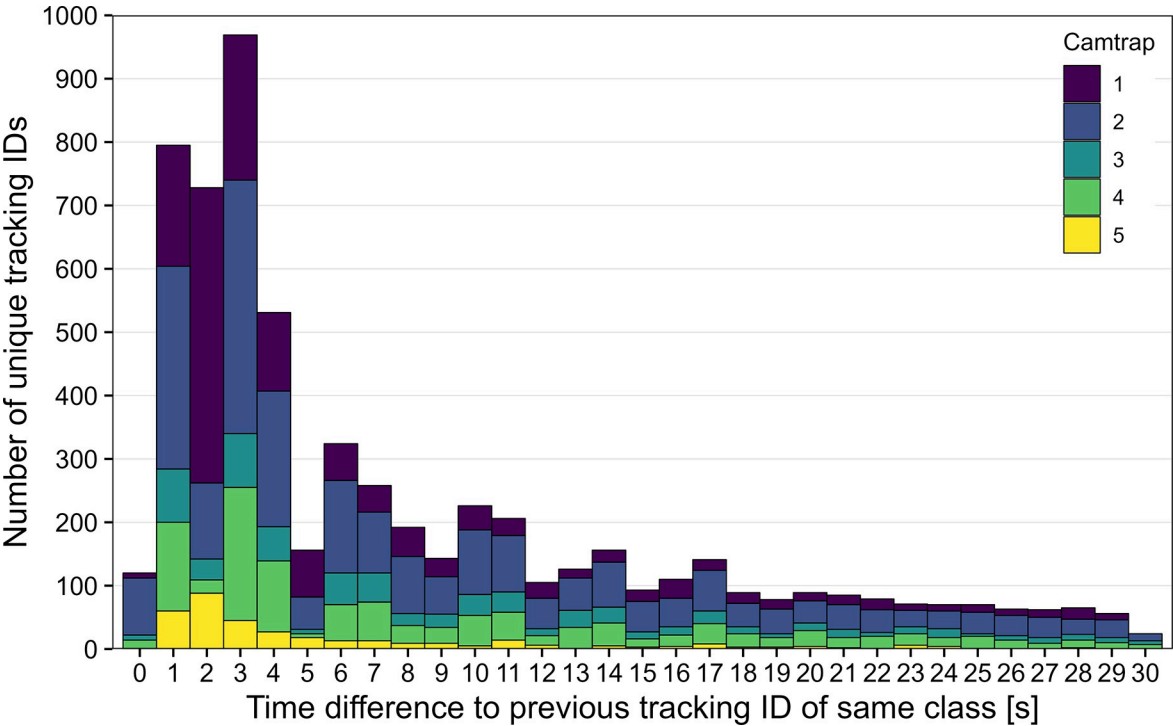

**Fig 13. Time difference to previous tracking ID classified as the same class.** Minimum time difference < 30 s of the previous five tracking IDs classified as the same class is shown. Data of non-insect classes not shown. All tracking IDs with less than three or more than 1,800 images were removed.

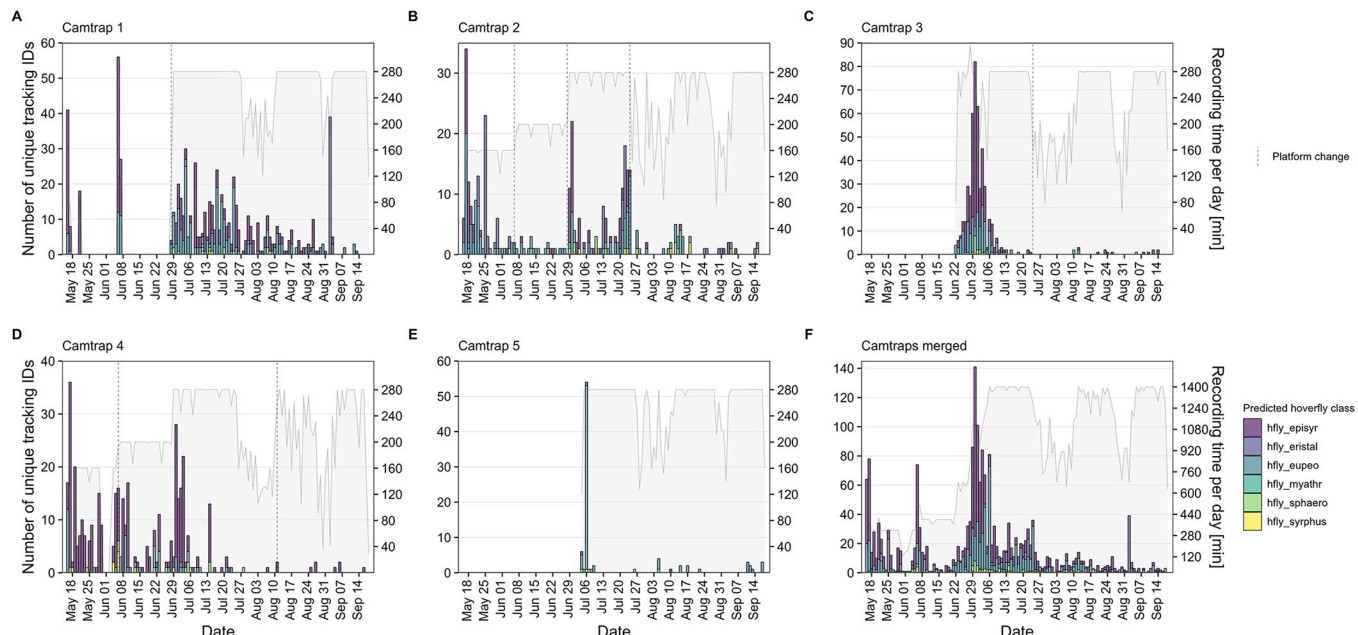

**Fig 14. Total number of unique tracking IDs classified as hoverfly and recording time per day.** (A–E) Data from camera traps 1–5. (F) Merged data from all camera traps. Grey lines/areas indicate the recording time per day. Dashed lines indicate a change of the flower platform. All tracking IDs with less than three or more than 1,800 images were removed.

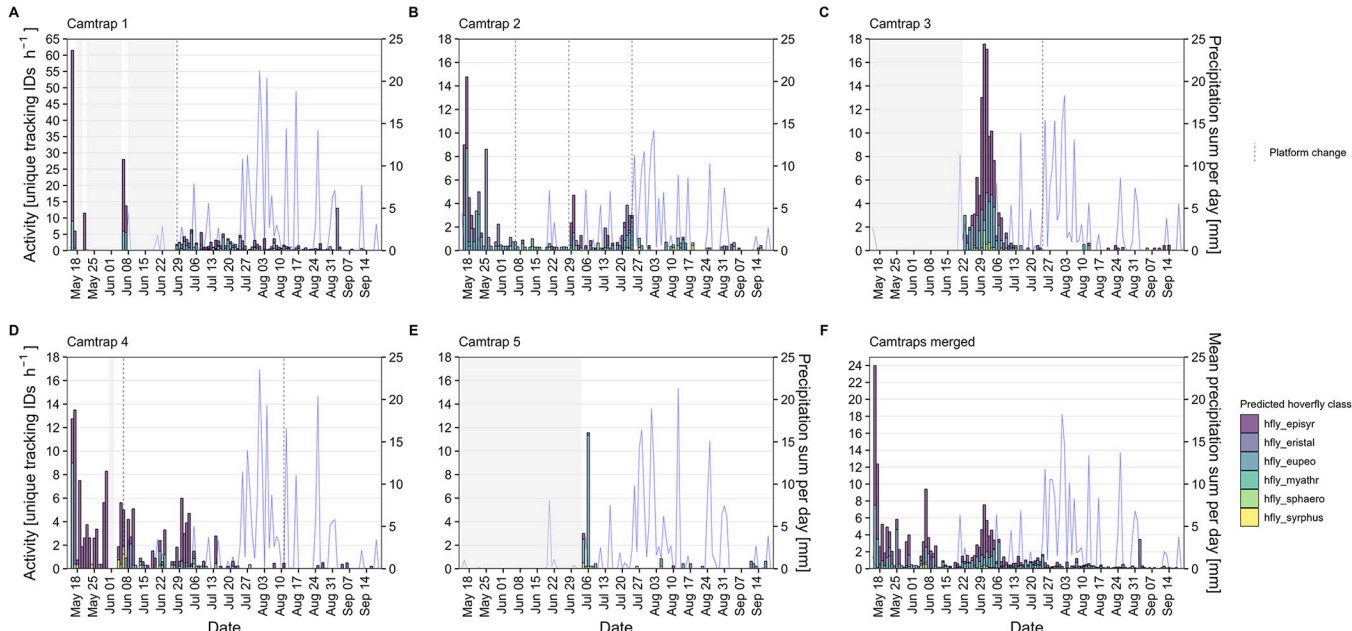

**Fig 15. Number of unique tracking IDs classified as hoverfly per hour and precipitation sum per day.** (A–E) Data from camera traps 1–5. (F) Merged data from all camera traps. Shaded areas indicate days without recordings. All tracking IDs with less than three or more than 1,800 images were removed. Weather data was taken from the nearest weather station (source: Deutscher Wetterdienst).

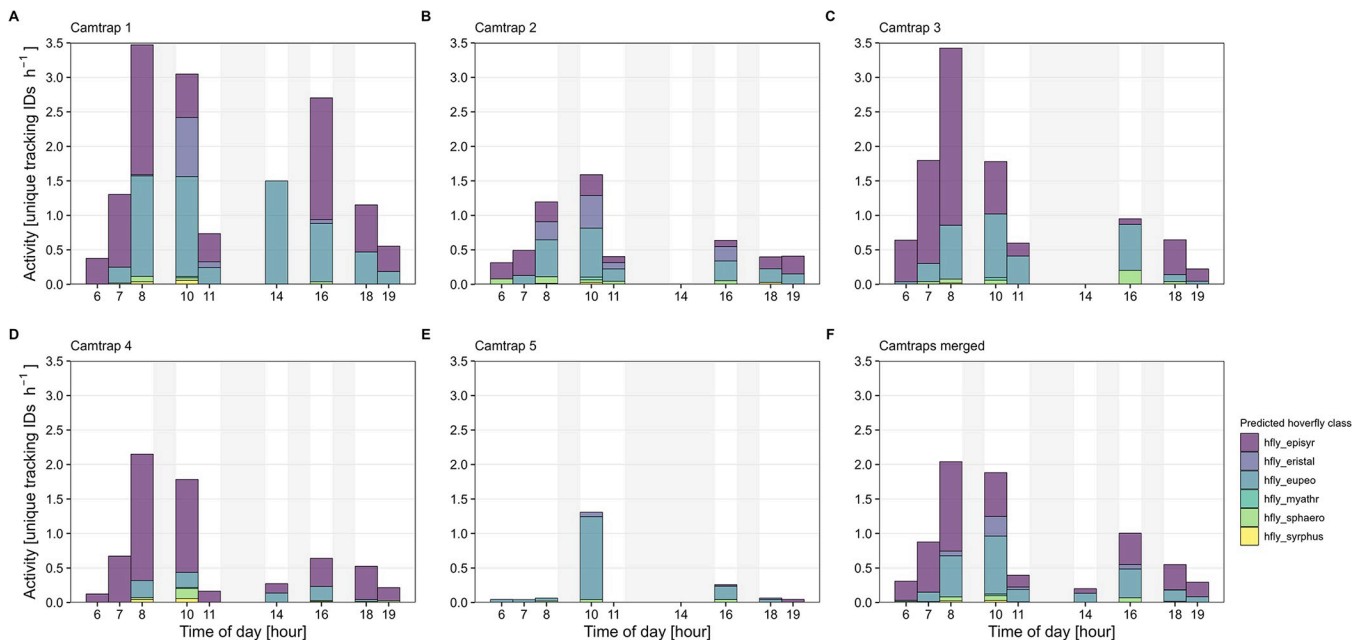

**Fig 16. Estimated hoverfly activity (number of unique tracking IDs per hour) per time of day.** (A–E) Data from camera traps 1–5. (F) Merged data from all camera traps. Shaded areas indicate hours without recordings. All tracking IDs with less than three or more than 1,800 images were removed.

## Discussion

The Insect Detect DIY camera trap system and associated software can be considered as an open-source development platform for anyone interested in automated insect monitoring, including non-professionals that are yet hesitant to delve into this rather complex topic. Due to its resistance to high temperatures and humidity, as well as a low power consumption of ~4.4 W combined with energy supplied by a solar panel, the camera trap system can be autonomously deployed during a whole season. We specifically target non-professionals as potential user group for deployment of the camera trap, such as citizen scientists involved in large-scale monitoring projects. Detailed step-by-step instructions for hardware assembly and software setup are provided at the corresponding documentation website [26] and allow everyone a full reproduction of the system with additional tips on software programming. The insect detection and classification models can be easily retrained on custom datasets without special hardware requirements by using the provided Google Colab notebooks [34].

The artificial flower platform, which is used as attractant for flower-visiting insects and background for the captured images, increases detection and tracking accuracy and can be standardized similar to yellow pan traps to compare insect populations at different sites. As for traditional traps that use visual features to attract insects, several characteristics, including the specific material, size, shape, color and orientation, affect the number and species assemblage of captured insects. Ongoing efforts are made to test and compare alternative materials, shapes and colors for the artificial flower platform. An updated platform design could further increase the visual attraction for a wider range of insect taxa or for specific target groups. By omitting the flower platform and training new detection models on appropriate datasets, the camera trap system could be adapted to different use cases, e.g. to monitor insects visiting real flowers, ground-dwelling arthropods (platform on ground) or nocturnal insects (vertical setup with light source).

### Processing pipeline

We implemented a two-stage approach that combines on-device insect detection and tracking in real time on low-resolution frames in the first stage (~12.5 fps for 1080p HQ frame resolution), with classification of the insect images cropped from synchronized high-resolution frames in the second stage. By using our proposed processing pipeline running on the camera trap hardware, it is sufficient to only store the cropped insect detections (~5–10 KB per image at 1080p resolution). This can save a significant amount of space on the microSD card, compared to storing the full frame (~0.5–1 MB per image at 1080p resolution). The separation of both steps also simplifies dataset management and encourages retraining of the classification model, as sorting the cropped insect images into class folders is much easier and faster compared to bounding box annotations required to train a new detection model. Cases with a low detection confidence and images that were classified to the wrong class or with a low probability (e.g. new insect species) should be identified, relabeled/resorted and added to a new dataset version to retrain the detection and/or classification model incrementally. Through this iterative active learning loop, the accuracy of the deployed models can be increased over time and will adapt to specific use cases and/or environments [47].

### Insect detection and tracking

While the provided insect detection models will generalize well for different homogeneous backgrounds and can also detect insect taxa not included in the training dataset, accuracy will drop significantly when using them with complex and dynamic backgrounds, such as natural vegetation. By utilizing datasets with annotated images that include these complex

backgrounds, models can also be trained to detect and classify insects under more challenging conditions [48–51]. Additionally, other techniques can be used to increase the detection and potentially also tracking accuracy of small insects in complex scenes, e.g. by using motion-informed enhancement of the images prior to insect detection [49, 51]. In general, the dataset size required to train models that can detect insects with a high accuracy increases with a higher visual variety of the insects and the background. If deployed in new environments, species that the model was not trained on might not be reliably detected (false negatives), which can lead to an underestimation of insect activity/abundance. Larger datasets with a high diversity of annotated insects on various backgrounds will increase overall model performance in the future.

The accuracy of the implemented object tracker depends on the frequency of received bounding box coordinates to calculate the object's trajectory and thereby the inference speed of the detection model. Fast and/or erratically moving insects, as well as insects coming very close to each other, can result in "jumping" tracking IDs and multiple counting of the same individuals. We show that the selection of different post-processing settings regarding the exclusion of tracking IDs with less than a specified number of images can result in more precise activity and abundance estimations. However, it is important to note that we only compared the final number of unique tracking IDs to the true frame visits in the presented experiment, without analyzing the number of false positive and false negative tracklets. As these could cancel each other out, our presented results can only estimate the tracking accuracy. Also, the synchronization with 1080p HQ frames could lead to different results, due to a faster pipeline speed (~12.5 fps) compared to the 4K HQ frame synchronization used in the experiment (~3.4 fps). Furthermore, the software of the OAK-1 device includes three additional object tracker types that can additionally influence the overall tracking accuracy. Further experiments with other insect species under field conditions and more detailed analyses comparing different settings during on-device processing and post-processing are necessary to validate the activity/abundance estimation by using the number of unique tracking IDs. Implementation of additional post-processing options, e.g. by incorporating temporal (timestamp) and spatial (insect coordinates) information from the captured metadata could furthermore increase the accuracy of the tracking results. More sophisticated tracking methods are available in other systems, e.g. a video-based processing pipeline that is able to detect, track and classify insects in heterogeneous environments of natural vegetation and enables the automated analysis of pollination events [52].

## Insect classification

When using our provided insect classification model for images of insect species that were not included in the training dataset, wrong classification results must be expected. New approaches incorporate the taxonomic hierarchy into the classification process to identify the lowest taxonomic rank for which a reliable classification result is achieved [53, 54]. Thereby, also completely new insect species that were not included in the training dataset can be correctly classified to higher taxonomic ranks (e.g. family or order), if they are morphologically similar to other species in the respective rank, which were included in the dataset.

When only images are used for automated classification, it is currently not possible to identify many insects to species level, as often microscopic structures have to be examined to distinguish closely related species. New approaches could fuse data generated by different sensors, such as images produced by an image sensor and wing beat frequency data generated by an opto-electronic sensor, to be able to identify insect species that could not be distinguished from similar species with images only [55].

## Insect monitoring data

In ecological insect surveys, traditional monitoring methods (e.g. Malaise traps or yellow pan traps) are usually deployed to lethally capture insect specimens that are then identified by trained experts to species level. Due to the high effort and cost required, these methods are often deployed in very limited timeframes and interpretation of the resulting data is restricted by its low temporal resolution. In contrast, data captured with automated methods is available at a significantly higher temporal resolution of up to several hours for each day and covers the whole season. Although the taxonomic resolution is currently still low for the automated classification of many insect taxa, combining data from both automated and traditional methods would benefit analyses significantly and widen the scope of possible interpretations [7]. At the same time, data on estimated insect abundance extracted from the camera trap metadata could be systematically compared with abundances acquired from traditional methods to calculate a conversion factor to allow for a more direct comparison of insect data captured with both methods.

The insect monitoring data presented in this paper can be seen as a proof of concept for automated monitoring with our proposed camera trap system. In this example, we mainly focus on hoverflies as target species, as this group provides important ecosystem services, such as pollination and natural pest control [56]. With camera trap data on the estimated daily activity of different hoverfly species, phenological information on different temporal scales can be extracted. This information could be used to e.g. study the impact of climate change on the phenology of specific species [57]. Activity changes during the day can also be investigated, e.g. to determine appropriate times for pesticide application with reduced impact on beneficial insects [58].

By changing the trap design and/or by adding olfactory attractants, such as pheromone dispensers, the camera trap system could be adapted to a more targeted monitoring of specific pest or invasive insect species [59, 60]. This also opens the opportunity to simultaneously monitor beneficial and pest insect species with the potential to significantly reduce pesticide use by facilitating data-informed decision making.

## Future perspectives

With rapid improvements in AI-based software and hardware capabilities, a broad application of these technologies for the next generation of biodiversity monitoring tools can be expected in the near future [7, 9, 19]. However, this also means that it is currently difficult to maintain a methodological continuity over longer periods, which is crucial for standardized long-term monitoring of insects. Storing the raw data generated by automated monitoring devices together with detailed metadata [61], including in-depth descriptions of the deployed trap design, enables reproduction and reprocessing of this data and will allow a comparison to updated technologies in the future [62].

Contrary to traditional methods, automated camera traps generate a permanent visual record of insects in a given location and time that can be reanalyzed with enhanced software capabilities in the future. As it is crucial to collect as many observations as possible to identify long-term trends and causes of insect population changes, future researchers will be grateful for every available record from the past that is our present now [63].

## Supporting information

**S1 Table. Description of the 27 classes from the image dataset that was used to train the insect classification model.** The images were sorted to the respective class by considering taxonomic and visual distinctions. In some cases, clear taxonomical separations are difficult from images only and the decision to sort an image to the respective class was based more on visual distinction. (DOCX)

**S2 Table. Comparison of different classification model architectures and hyperparameter settings supported by YOLOv5 classification model training.** All models were trained on a custom dataset with 21,000 images (14,686 in train split) and default hyperparameters. Metrics are shown on the dataset validation split (4,189 images) and dataset test split (2,125 images) for the converted models in ONNX format.
(DOCX)

**S3 Table. Metrics of the EfficientNet-B0 insect classification model, validated on a real-world dataset.** The model was trained on a custom dataset with 21,000 images (14,686 in train split), scaled to 128x128 pixel, to 20 epochs with batch size 64 and default hyperparameters. Metrics are shown on a real-world dataset (97,671 images) for the converted model in ONNX format. All images were classified and subsequently verified and sorted to the correct class in the case of a wrong classification by the model. A dummy image was added for each of the classes "bug_grapho" and "fly_empi", as no images of both were captured during the deployment period. Results for both classes must be ignored.
(DOCX)

**S1 Fig. Hardware schematic of the electronic camera trap components.** Nominal voltage is shown for the PiJuice 12,000 mAh battery.
(TIFF)

**S2 Fig. Examples for limitations in the detection and tracking accuracy.** (A) The same tracking ID is assigned to insects coming close to each other. (B) A fast-moving insect is not correctly tracked, with the risk of a new tracking ID being assigned to the same individual.
(TIFF)

**S3 Fig. Total number of unique tracking IDs for each predicted class.** Merged data from all five camera traps deployed from mid-May to mid-September 2023. All tracking IDs with less than three or more than 1,800 images were removed.
(TIFF)

**S4 Fig. Total recording time per time of day.** Merged data from all five camera traps deployed from mid-May to mid-September 2023.
(TIFF)

**S5 Fig. Estimated bee activity (number of unique tracking IDs per hour) per time of day.** Merged data from all five camera traps deployed from mid-May to mid-September 2023. Shaded areas indicate hours without recordings. All tracking IDs with less than three or more than 1,800 images were removed.
(TIFF)

**S6 Fig. Estimated fly activity (number of unique tracking IDs per hour) per time of day.** Merged data from all five camera traps deployed from mid-May to mid-September 2023. Shaded areas indicate hours without recordings. All tracking IDs with less than three or more than 1,800 images were removed.
(TIFF)

## Acknowledgments

We would like to thank all caretakers and owners of the orchard meadows for letting us deploy the camera traps during the whole season. Thanks a lot to everybody in the automated insect monitoring community for constant feedback and support. This study is part of the joint

project "National Monitoring of Biodiversity in Agricultural Landscapes" (MonViA) of the German Federal Ministry of Food and Agriculture.

## Author Contributions

**Conceptualization:** Maximilian Sittinger, Johannes Uhler, Maximilian Pink, Annette Herz.

**Data curation:** Maximilian Sittinger, Johannes Uhler, Maximilian Pink.

**Formal analysis:** Maximilian Sittinger.

**Funding acquisition:** Annette Herz.

**Investigation:** Maximilian Sittinger, Johannes Uhler, Maximilian Pink.

**Methodology:** Maximilian Sittinger.

**Project administration:** Annette Herz.

**Software:** Maximilian Sittinger.

**Supervision:** Annette Herz.

**Validation:** Maximilian Sittinger, Johannes Uhler, Maximilian Pink.

**Visualization:** Maximilian Sittinger.

**Writing – original draft:** Maximilian Sittinger.

**Writing – review & editing:** Maximilian Sittinger, Johannes Uhler, Maximilian Pink, Annette Herz.

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
