## [Decision Letter · Decision Letter 0]

26 Dec 2023

PONE-D-23-38812Insect Detect: An open-source DIY camera trap for automated insect monitoringPLOS ONE

Dear Dr. Sittinger,

Thank you for submitting your manuscript to PLOS ONE. After careful consideration, we feel that it has merit but does not fully meet PLOS ONE’s publication criteria as it currently stands. Therefore, we invite you to submit a revised version of the manuscript that addresses the points raised during the review process.

We look forward to receiving your revised manuscript.

Kind regards,

Ramzi Mansour

Academic Editor

PLOS ONE

Journal Requirements:

Reviewers' comments:

Reviewer's Responses to Questions

**Comments to the Author**

1. Is the manuscript technically sound, and do the data support the conclusions?

Reviewer #1: Yes

Reviewer #2: Partly

2. Has the statistical analysis been performed appropriately and rigorously? 

Reviewer #1: Yes

Reviewer #2: N/A

3. Have the authors made all data underlying the findings in their manuscript fully available?

Reviewer #1: Yes

Reviewer #2: Yes

4. Is the manuscript presented in an intelligible fashion and written in standard English?

Reviewer #1: Yes

Reviewer #2: Yes

5. Review Comments to the Author

Reviewer #1: 

Comments to the Authors:

The aim of this paper is to present the Insect Detect DIY camera trap, a low-cost and customizable automated monitoring system for flower-visiting insects, utilizing off-the-shelf hardware and open-source software, allowing for large data collection and reliable insect activity estimation. The paper provides substantial value to the scientific community, as it addresses the current hot topic of obtaining reliable abundance counts of insects, which holds crucial importance for biodiversity conservation. However, the readability could be enhanced, particularly concerning the main goal of the study, which revolves around the detection, tracking, classification algorithms, and their corresponding results.

For better understanding of the study, it would be helpful to clarify the points below.

1. Consider using full names Bombus, Coccinellidae, Coleoptera, Graphosoma etc. for Figures, Tables and within the text, as it improves the overall readability of the study instead of “bee_bombus”, “beetle_cocci”, “bug”, “bug_grapho”, etc.

2. The frequent mention of Python scripts, R-scripts, or output files using their specific output format (.csv) in the text can be reduced to enhance readability. Instead, it is recommended to use a simpler phrase such as "The data output for automated insect monitoring" rather than specifying "The data output from the Python script for automated insect monitoring."

3. Similar to the statement in line 442-444, "with the default settings (“-min_tracks 3” and “-max_tracks 1800”) to exclude all tracking IDs with less than three or more than 1,800 images," there are instances where the text reads like camera trap documentation, which is somewhat understandable. However, a paper presenting impressive results like this should strive for good readability.

4. The paper includes a valuable GitHub repository, which is highly commendable. Therefore, it is suggested to remove all comments regarding code or scripts from the manuscript, including the entire section 2.4. Data Visualization, to enhance readability.

5. I hope I didn't miss it, but I was curious why the focus was solely on hoverflies in the results and particularly in the discussion, considering that the camera and model classified other intriguing insects as well. It may be beneficial to address this in the introduction, as it was rather surprising.

6. Starting at Fig. 6 (Fig 8., S2 Table, line 274) the text “The model was trained on a custom dataset with 21,000” images is repeated four times throughout the manuscript. I think it is enough to mentioned it once in detail and the other times in a shortened version.

7. Fig. 2 is highly impressive. I would suggest that, since it falls under the processing pipeline section, some essential details about the YOLO detection model and its hyperparameters could be included. Additionally, it might be beneficial to incorporate information about the post-processing classification stage, the Efficientnet, and a few minor but significant hyperparameters.

8. If I understood correctly, the YOLO models for insect detection were trained on 1335 images for 300 epochs, while the Efficientnet model for insect classification was trained on 21000 images for 20 epochs. It would be helpful if you could provide a clearer explanation for this choice of experimental setup, particularly addressing and discussing any concerns regarding potential overfitting when training with just 1335 images for 300 epochs.

9. Additionally, it could be beneficial to mention that Resnet50, YOLO standard backbone, and Efficientnet were employed in the study, with the final decision to use Efficientnet based on its superior performance.

10. Caption for Fig. 8 is too lengthy; consider providing a more detailed explanation in Section 3.2 on Insect Classification Validation.

Reviewer #2: 

This paper presents an automated, do-it-yourself camera trap system for monitoring flower-visiting insects. The system consists of two components: a real-time camera with a deep learning-based object detector that identifies and captures insect images on an artificial platform, and an insect classification model that identifies species from captured images.

The camera trap's accuracy was tested in a controlled laboratory experiment using hoverflies as a test species. The classification model was validated using images captured by deploying the camera trap at test sites. Additionally, the authors conducted a brief case study demonstrating the system's capabilities by analyzing data related to hoverfly behavior.

The authors employed appropriate methods and provided detailed documentation and guidance to ensure the camera trap's reproducibility for non-expert users, encouraging citizen science engagement in insect monitoring. This is a significant contribution of the paper.

However, I believe some issues related to the camera trap's evaluation should be addressed before accepting the paper for publication. If these concerns can be satisfactorily addressed, I recommend accepting the paper for publication in PLoS ONE.

Major comments

(1) A key strength of this paper, compared to the existing literature discussed in the introduction, lies in its development of a camera trap system built with open-source software. This system, accessible to non-experts in computer science or engineering, addresses a significant gap in the field. However, the introduction currently lacks a clear explanation of this gap and its significance. To provide readers with better understanding of this paper's contributions, I recommend the authors include a concise description of this research gap before introducing current research at Line 93.

(2) While the introduction mentions real-time processing, it doesn't fully justify its preference over offline processing. Providing more detail on this aspect would benefit the reader's understanding of the proposed system's necessity.

(3) The abstract (Line 34) and Discussion section (Line 587) states that "on-device detection and tracking reliably estimated insect activity/abundance...". I am a bit confused about this statement as under Insect track evaluation results (Line 390) I did not find any quantitative metrics that was calculated to measure the reliability or accuracy of insect tracking to back up the above mentioned statement. While there are standard methods to evaluate accuracy of a Multi-object tracking problem (e.g Luiten et al. "HOTA: A Higher Order Metric for Evaluating Multi-Object Tracking", IJCV 2020.), I do not think a complete evaluation of tracking is necessary as the final measurement of the system is the number of insects landed on the platform. However, as the reliability and accuracy of insect counting is integral for the system, some sort of quantitative metric is necessary.

Authors have attempted to address this by comparing the total number of automatically detected insect tracks to those observed visually. I do not agree this is an appropriate way to measure the reliability because the metric itself is susceptible to misleading results due to potential cancellation of false positives and negatives.

Alternatively, I recommend presenting separate counts for True Positives, False Positives, and False Negatives generated by the on-device tracking. These counts can then be used to calculate Precision, Recall, and F-score for the camera trap's detection of hoverflies (https://en.wikipedia.org/wiki/Precision_and_recall). Similar metrics have been used by other research cited in this paper (e.g: 17, 48). Presenting these final average metrics (Precision, Recall, F-score) in the abstract and Discussion/Conclusion sections would provide a more quantitative and reliable measure of the system's detection accuracy.

(4) In the Introduction (Line 102), Methods (Line 191) and Discussion (Line 588) authors state that the “...the whole pipeline runs at 190 ~12.5 fps (1080p HQ frame resolution), which is sufficient to reliably track most insects…”. However, the evaluations presented in the insect tracking section (Line 320) utilize 4K HQ frames. This inconsistency raises concerns about which resolution was used for the actual classification task, as the authors mention both 1080p and 4K image synchronization throughout the manuscript. Since image resolution can significantly impact classification accuracy, it's crucial to clarify this point. If the camera trap system and its classification model were indeed tested on 4K captured (and cropped) images, reporting the processing speed for 4K frames would be more accurate and reflect the true operational speed of the system. Additionally, a brief discussion on the trade-offs between speed and resolution, and how these choices affect the monitoring of different insect species, would be valuable for readers.

(5) I appreciate the detailed hardware assembly instructions provided in the documentation. However, it's currently unclear how the individual components connect to each other. To address this, I suggest incorporating a simple hardware schematic (complimenting Figure 1) to visually illustrate the connections between each component.

(6) Using artificial flower visits as a proxy for insect counts may not be as accurate as direct flower observations. This is because insect visitation to artificial flowers can be influenced by various characteristics of the flowers themselves. Please include a brief discussion on how the choice of platform materials and colors could affect capture rates, thereby improving the reliability of this method.

(7) Line 87-89: The detection rate and accuracy of a deep learning model depends on various factors including its architecture and quality of the training dataset. Hence, the use of deep learning-based models for insect detection can result in false negative detections leading to underestimating insect counts. Please briefly mention this in the introduction and further discuss how this drawback could be overcome in the Discussion section.

Minor Comments

Presentation Structure: Authors have presented this study in easy to understand clear language. However, the structure the manuscript presented was confusing to me. I would like to propose authors to consider restructuring the manuscripts Methods and Materials and Results sections. Here Methods and Materials sections would contain two subsections on Camera Trap and Insect Classification model and only contain methodology associated with it. The Results section (or Experiments and Results) sections would contain all the experimental evaluations including results of YOLO models, classification model, field deployment etc, divided among two subsections on Camera Trap and Insect Classification model.

Line 24: Not all traditional monitoring methods (e.g. focal flower observations or quadrat observations, transect walks) may provide data with high taxonomic resolution.

Line 68 and 71: Other research that uses motion detection for insect monitoring include “van der Voort, Genevieve E., et al. "Continuous video capture, and pollinia tracking, in Platanthera (Orchidaceae) reveal new insect visitors and potential pollinators." PeerJ 10 (2022): e13191.”, “Steen, Ronny. "Diel activity, frequency and visit duration of pollinators in focal plants: in situ automatic camera monitoring and data processing." Methods in Ecology and Evolution 8.2 (2017): 203-213.”

Line 106: Please provide an appropriate reference and data on the speed of the hoverfly species.

Line 123: Is 91 Wh the combined capacity of the two batteries?

Line 127: What are the dimensions of the platform?

Line 131: I suggest including the component list and associated cost values also in the supporting materials. This is because the cost provided in the manuscript may change over time.

Line 158: It is unclear on what types other homogeneous backgrounds the YOLO model was tested with. Could you please clarify?

Line 161: Why were the metrics not calculated for the test split?

Line 178: Please provide references for Kalman Filter and Hungarian Algorithm.

Line 191: [See Major comment 3 and 4]

Line 199: Please include the type of metadata recorded in the figure caption.

Line 205: Please include more information on how the power consumption was measured. What device did you use to measure the energy consumption? Under what ambient conditions (temperature, humidity) was the test conducted? Was the solar panel connected to the system during this test? Were 5 insects tracked simultaneously or sequentially during the test? Also, what was the camera resolution?

Line 208: Was the estimate of 20 hours calculated considering the threshold value mentioned in line 210?

Line 229: Could you please explain why YOLOv5 was used on captured images first to classification without directly using EfficientNet-B0 on captured images?

Line 234: Please provide a reference to the EfficientNet-B0 model.

Line 259: Could you please explain why high inference speed is critical for this step. As the images are classified offline and as the main aim is to achieve the best classification accuracy, shouldn’t the accuracy be prioritized over inference speed?

Line 313: Please change “mm” to millimeters.

Line 324: Please provide the video settings used by the smartphone camera including its resolution and framerate. Also, could you please provide more detail on the speed the smartphone camera video was played at? (e.g 50% of original speed).

Line 364: Please explain why a threshold of 70% was used? Why not set a lower value allowing us to record more data?

Line 416: I suggest authors include the S3 table in the manuscript as it reflects the performance of the classification model on the real world data. Also Table 2 can be moved to Supplementary materials.

Line 516: I suggest authors present the analysis of hoverfly behavior under a subsection “Example data analysis" or " Case Study”.

Line 579: Could you please provide any references to support the statement that the artificial flower platform can be standardized similarly to yellow pan traps. [Also see Major comment 6].

Line 582: Please change the “camera trap system” to “camera trap hardware” as the software system was not evaluated for monitoring insects visiting real flowers or in outdoor settings.

Line 583: I agree with authors that the presented hardware system can be used to monitor insects visiting real flowers. However, it is unclear how the software solution will translate to monitoring insect visits to real flowers. Basic monitoring insect visit to real flowers requires detecting insects in an image, obtaining its coordinates, tracking its position and movements with changes in environment and its posture, and comparing insect coordinates with flower coordinates to identify flower visits. Could you please mention how the presented methods can achieve these requirements, or alternatively present these requirements as future work. This could be an expansion to the discussion presented under Insect detection and tracking subsection in the Discussion section.

Line 603: Under insect detection and tracking section please include a discussion on how the methods presented in this study can be implemented with different IoT platforms and how development of more efficient computational platforms can leverage the results of this study.

Line 609, 611, 625: Studies cited in [45,46,48] can also be discussed in the introduction section.

Line 611: Other research that used motion enhancement includes “Ratnayake, Malika Nisal, Adrian G. Dyer, and Alan Dorin. "Tracking individual honeybees among wildflower clusters with computer vision-facilitated pollinator monitoring." Plos one 16.2 (2021): e0239504.”

Line 620 - 622: Currently there is research being conducted on re-identification of insects (see. Borlinghaus, Parzival, Frederic Tausch, and Luca Rettenberger. "A Purely Visual Re-ID Approach for Bumblebees (Bombus terrestris)." Smart Agricultural Technology 3 (2023): 100135.”). Please discuss the possibility of using a similar mechanism for the proposed study to improve its sampling accuracy.

Fig 2.: Please label the purple line from Script node to cropped detections.

Fig 4: Please indicate the start and end of the recording period.

Fig 6 and Fig 8: Please label the color bar.

Fig. 7: I suggest removing the “15 min recording” from axis titles to simplify the plot. Also rename y axis to “Ground truth” or “Manual video observations” and the x axis to “Camera trap recordings”.

Fig 11. I suggest moving this figure to supplementary materials. Results presented in 11F are a bit confusing as all the camera traps were not deployed for the same period of time. I recommend removing plot 11F. Could you please provide more information on how sunshine was measured? Please adjust the secondary y axis scale to match that of the primary y axis. Also include a legend describing what each line in the plot represents.

Fig.12: Please make all y axis scales 12A-12E same to enable easy comparison of data across camera traps.

Fig 14: As the recording time per day varies between camera traps, I suggest normalizing the number of unique tracking IDs by the recording time. Please make all y axis scales 14A-14E same to enable easy comparison of data across camera traps

Fig 15: As the discussion does not extensively analyze the relationship between rainfall and hoverfly activity, this Figure can be removed.

6. PLOS authors have the option to publish the peer review history of their article (what does this mean?). If published, this will include your full peer review and any attached files.

Reviewer #1: No

Reviewer #2: No

---

## [Author Response · Author response to Decision Letter 0]

29 Jan 2024

Reviewer #1:

Comments to the Authors:

The aim of this paper is to present the Insect Detect DIY camera trap, a low-cost and customizable automated monitoring system for flower-visiting insects, utilizing off-the-shelf hardware and open-source software, allowing for large data collection and reliable insect activity estimation. The paper provides substantial value to the scientific community, as it addresses the current hot topic of obtaining reliable abundance counts of insects, which holds crucial importance for biodiversity conservation. However, the readability could be enhanced, particularly concerning the main goal of the study, which revolves around the detection, tracking, classification algorithms, and their corresponding results.

For better understanding of the study, it would be helpful to clarify the points below.

1. Consider using full names Bombus, Coccinellidae, Coleoptera, Graphosoma etc. for Figures, Tables and within the text, as it improves the overall readability of the study instead of “bee_bombus”, “beetle_cocci”, “bug”, “bug_grapho”, etc.

Authors response: Thank you for your suggestion. We agree that using the full taxonomic names could increase the readability compared to using the class labels with sometimes abbreviated taxonomic names. However, we also do not want readers to assume that our presented classification model is able to precisely separate insects into their respective taxonomic ranks. To avoid this assumption, we purposely used the class labels in the Materials and methods section when describing the classes, as well as in the figures and tables. In lines 280-281 we refer to S1 table, in which all class labels are described and associated to their respective taxa(-groups). Additionally, we added some more information in lines 280-281 to make it clear that not all classes correspond to strictly defined insect taxa. In the Results section we mostly used the full names together with the class labels to improve readability. In line 532, we added “[…] wild bees excluding Bombus sp. (“bee”) […]” to make the distinction to the two other classes containing bee species (Apis mellifera and Bombus sp.) clearer.

2. The frequent mention of Python scripts, R-scripts, or output files using their specific output format (.csv) in the text can be reduced to enhance readability. Instead, it is recommended to use a simpler phrase such as "The data output for automated insect monitoring" rather than specifying "The data output from the Python script for automated insect monitoring."

Authors response: Thank you for this reasonable suggestion. As recommended, we simplified the phrasing in lines 258, 264, 321, 323, 328 and 423-424.

3. Similar to the statement in line 442-444, "with the default settings (“-min_tracks 3” and “-max_tracks 1800”) to exclude all tracking IDs with less than three or more than 1,800 images," there are instances where the text reads like camera trap documentation, which is somewhat understandable. However, a paper presenting impressive results like this should strive for good readability.

Authors response: Thanks a lot for this nice comment. As recommended, we simplified the phrasing in lines 367 and 485. There are still more instances in the text that might read like camera trap documentation, but this is due to the scope of the manuscript describing a novel methodology/tool. We tried to simplify as much as possible, but in our opinion many aspects of the hardware/software must be described with some minimum amount of detail to be able to fully understand the whole system.

4. The paper includes a valuable GitHub repository, which is highly commendable. Therefore, it is suggested to remove all comments regarding code or scripts from the manuscript, including the entire section 2.4. Data Visualization, to enhance readability.

Authors response: Thank you for this suggestion. However, we think that the details we mention about our code are necessary to understand the respective context. Therefore, we would also like to keep the last section in Material and methods “Data visualization”, as it describes very briefly how the analyses and plots shown in the Results section were created. We also think that this short section is the best place to refer to the Zenodo repository where all R scripts and associated data are published to reproduce and check our analyses.

5. I hope I didn't miss it, but I was curious why the focus was solely on hoverflies in the results and particularly in the discussion, considering that the camera and model classified other intriguing insects as well. It may be beneficial to address this in the introduction, as it was rather surprising.

Authors response: Thank you for this question. We do address this briefly in the manuscript. In lines 558-559, we mention that “Data from the images classified as one of the hoverfly classes is presented in more detail in the following plots, as an example for a functional group of special interest.” In lines 722-724 we explain that “In this example, we mainly focus on hoverflies as target species, as this group provides important ecosystem services, such as pollination and natural pest control”. As many hoverfly species combine two important ecosystem functions in the agricultural landscape (adults: pollination; larvae: pest control) and we trained our classification model to differentiate six different hoverfly species(-groups), we chose to focus on this group for the proof-of-concept data analysis. The tracking experiment was carried out with the hoverfly species Episyrphus balteatus, as we could rear this species in our facilities. This species and relatives are known to fly rather fast and erratically, which made it suitable to test the tracking abilities under more difficult conditions. In our opinion, we made it clear that this species group was only shown as a proof of concept and other groups could be captured and analyzed in a similar way (e.g. with updated classification models that can differentiate more classes). As recommended, we added more information about our focus on hoverflies in the introduction in line 126: “[…] with a focus on six different hoverfly species(-groups).”

6. Starting at Fig. 6 (Fig 8., S2 Table, line 274) the text “The model was trained on a custom dataset with 21,000” images is repeated four times throughout the manuscript. I think it is enough to mentioned it once in detail and the other times in a shortened version.

Authors response: As recommended, we shortened this information in lines 305-306, 311-312 and 471-472. For S2 Table we did not shorten the text, as it could also be viewed independently from the main text and already includes the minimum amount of detail that is necessary to accompany the table content.

7. Fig. 2 is highly impressive. I would suggest that, since it falls under the processing pipeline section, some essential details about the YOLO detection model and its hyperparameters could be included. Additionally, it might be beneficial to incorporate information about the post-processing classification stage, the Efficientnet, and a few minor but significant hyperparameters.

Authors response: Thank you very much for this nice comment and the suggestion. We understand that including more details in Fig 2 could give the reader more information about the whole processing pipeline. However, we also want to keep the diagram as simple and general as possible, to facilitate understanding for non-professionals as well. As the detection models and their respective hyperparameters can be freely customized by each user (e.g. YOLOv5/v6/v7/v8 but also other architectures are possible), while still running them in this same processing pipeline, including more detailed information could potentially give the false impression that only this one specific model can be used in the pipeline. The diagram in Fig 2 only shows the processing pipeline that is run in real time on the camera trap hardware. We clearly mention in the manuscript text, that the captured images are classified in a subsequent step and metadata should be post-processed prior to further analysis.

8. If I understood correctly, the YOLO models for insect detection were trained on 1335 images for 300 epochs, while the Efficientnet model for insect classification was trained on 21000 images for 20 epochs. It would be helpful if you could provide a clearer explanation for this choice of experimental setup, particularly addressing and discussing any concerns regarding potential overfitting when training with just 1335 images for 300 epochs.

Authors response: We purposely omitted specific details about the training runs leading up to the models that we published, as this would go beyond the scope of this paper. By publishing all datasets and Google Colab notebooks that were used to train the presented models, we encourage everyone interested to reproduce our training results and test e.g. different hyperparameters such as the number of epochs the model is trained to. While training the models we considered the best practices to avoid overfitting, e.g. by observing the loss on the validation set. Training to 300 epochs for the YOLO detection models and to 20 epochs for the EfficientNet-B0 classification model resulted in the highest possible mAP/top-1 accuracy respectively without an increase in the validation loss. Strictly speaking, the detection models probably kind of overfit to the flower platform, which always provides the same background. But this is expected and we make it clear that our provided models will only work well if the background for insect detection is constant and homogeneous (e.g. in lines 652-654).

9. Additionally, it could be beneficial to mention that Resnet50, YOLO standard backbone, and Efficientnet were employed in the study, with the final decision to use Efficientnet based on its superior performance.

Authors response: Thanks a lot for this suggestion. We added this information in line 291, details on the model comparison can be found in S2 Table.

10. Caption for Fig. 8 is too lengthy; consider providing a more detailed explanation in Section 3.2 on Insect Classification Validation.

Authors response: Thank you for this hint. We shortened the caption for Fig 8 in lines 471-472 to not include details about the classification model training, that were already provided previously in the text (see also response to comment 6).

Reviewer #2: 

This paper presents an automated, do-it-yourself camera trap system for monitoring flower-visiting insects. The system consists of two components: a real-time camera with a deep learning-based object detector that identifies and captures insect images on an artificial platform, and an insect classification model that identifies species from captured images.

The camera trap's accuracy was tested in a controlled laboratory experiment using hoverflies as a test species. The classification model was validated using images captured by deploying the camera trap at test sites. Additionally, the authors conducted a brief case study demonstrating the system's capabilities by analyzing data related to hoverfly behavior.

The authors employed appropriate methods and provided detailed documentation and guidance to ensure the camera trap's reproducibility for non-expert users, encouraging citizen science engagement in insect monitoring. This is a significant contribution of the paper.

However, I believe some issues related to the camera trap's evaluation should be addressed before accepting the paper for publication. If these concerns can be satisfactorily addressed, I recommend accepting the paper for publication in PLoS ONE.

Major comments

(1) A key strength of this paper, compared to the existing literature discussed in the introduction, lies in its development of a camera trap system built with open-source software. This system, accessible to non-experts in computer science or engineering, addresses a significant gap in the field. However, the introduction currently lacks a clear explanation of this gap and its significance. To provide readers with better understanding of this paper's contributions, I recommend the authors include a concise description of this research gap before introducing current research at Line 93.

Authors response: Thank you very much for this nice comment and suggestion. As recommended, we added a short section in lines 102-106 to emphasize the mentioned research gap regarding the accessibility of these systems to non-professionals. From our point of view the significance of this contribution is explicitly addressed in lines 108-110 with the sentence “Our goal was to develop a camera trap that could be easily utilized in monitoring projects involving citizen scientists to achieve a broader application potential.” We now also emphasize the accessibility to non-professionals more at the beginning of the Discussion, by adding “[…] including non-professionals that are yet hesitant to delve into this rather complex topic.” in line 610.

(2) While the introduction mentions real-time processing, it doesn't fully justify its preference over offline processing. Providing more detail on this aspect would benefit the reader's understanding of the proposed system's necessity.

Authors response: Thanks for this very reasonable suggestion. We agree that we did not give enough detail about this aspect and added more information about potential benefits of real-time on-device processing in lines 92-99.

(3) The abstract (Line 34) and Discussion section (Line 587) states that "on-device detection and tracking reliably estimated insect activity/abundance...". I am a bit confused about this statement as under Insect track evaluation results (Line 390) I did not find any quantitative metrics that was calculated to measure the reliability or accuracy of insect tracking to back up the above mentioned statement. While there are standard methods to evaluate accuracy of a Multi-object tracking problem (e.g Luiten et al. "HOTA: A Higher Order Metric for Evaluating Multi-Object Tracking", IJCV 2020.), I do not think a complete evaluation of tracking is necessary as the final measurement of the system is the number of insects landed on the platform. However, as the reliability and accuracy of insect counting is integral for the system, some sort of quantitative metric is necessary.

Authors have attempted to address this by comparing the total number of automatically detected insect tracks to those observed visually. I do not agree this is an appropriate way to measure the reliability because the metric itself is susceptible to misleading results due to potential cancellation of false positives and negatives.

Alternatively, I recommend presenting separate counts for True Positives, False Positives, and False Negatives generated by the on-device tracking. These counts can then be used to calculate Precision, Recall, and F-score for the camera trap's detection of hoverflies (https://en.wikipedia.org/wiki/Precision_and_recall). Similar metrics have been used by other research cited in this paper (e.g: 17, 48). Presenting these final average metrics (Precision, Recall, F-score) in the abstract and Discussion/Conclusion sections would provide a more quantitative and reliable measure of the system's detection accuracy.

Authors response: Thank you very much for this detailed comment and very valuable suggestion. We fully agree that the presented metric of comparing the true hoverfly platform/frame visits, manually counted in smartphone video recordings of the platform, with the number of unique tracking IDs, generated by the on-device tracking and subsequent post-processing with different settings, is susceptible to misleading results due to potential cancellation of false positives and false negatives. Presenting more established metrics, such as Precision, Recall and F1-score not only for the detection accuracy (calculated on the validation set and shown in Table 1) but also for the tracking accuracy, would indeed support our previous statement of “[…] reliably estimated insect activity/abundance […]” more profoundly. As the tracker algorithm directly uses the output of the detection model (bounding box coordinates), the presented detection accuracy can still be considered as a first estimation of also the tracking accuracy, e.g. regarding a high Recall of the detection model corresponding to a presumably lower number of false negative tracklets (= tracking ID) occurring during object tracking (but potentially higher number of false positives). From our experience, false negative tracklets, meaning that an insect is not detected when entering the frame and no bounding box coordinates are given to the object tracker, are very rare. On the other hand, the assignment of false positive tracklets (e.g. multiple tracking IDs for the same individual because it is moving too fast or swaps the ID with an individual coming to close) occur much more often, as can be seen in Fig 7 for the unprocessed data (“-min_tracks 1”). We are aware that this is only anecdotal evidence at this point, but still believe that the presented experiment can provide some interesting information about the performance of the on-device tracking.

Compared to the mentioned research of Bjerge et al. 2021 (Ref. 17) and Ratnayake et al. 2022 (Ref. 48), we had to deal with data that are more complicated to analyze regarding calculation of true/false positive and false negative tracklets. In contrast to both works, we did not have video/full frame recordings with associated metadata available to evaluate the number of true positive, false positive and false negative tracklets. With only cropped detections and metadata (including timestamp, tracking ID and bounding box coordinates) together with videos taken with a different camera while running the automated insect monitoring script, it is much more difficult to associate the automatically captured tracklets with the true frame visits, as no bounding boxes with tracking IDs can be drawn on the frames from the video recordings. This would have required a frame-by-frame analysis of the videos together with manually comparing the exact timestamps of every hoverfly tracklet (= tracking ID). Also, we would like to remark that Bjerge et al. 2021 (Ref. 17) did in fact not evaluate false negative tracklets in the sense of undetected insects, but only in the sense of wrongly classified species. Their presented metrics can therefore not be interpreted in a way the reviewer is indicating.

Furthermore, as we compare different post-processing settings regarding the minimum number of images (captured per second) that are required to include the respective tracking ID in the final data, the true/false positive and false negative tracklets would still not fully describe the reliability of the activity/abundance estimations. In the presented tracking experiment, we wanted to show an estimation of the tracking accuracy under difficult conditions. For this reason, we chose a fast and often erratically flying hoverfly species at a high activity level (due to containment in an insect cage at high numbers) together with 4K HQ frame synchronization for the on-device processing pipeline at a reduced speed of ~3.4 fps compared to ~12.5 fps for 1080p HQ frame synchronization. At a lower framerate the object tracker receives bounding box coordinates in a lower frequency, which means that the calculation of the object trajectory is less precise which can lead to “jumping” tracking IDs if the track of an individual is lost. This is another possible factor that will affect the overall tracking accuracy of the system, depending on the user settings. Also there are three more object tracking types available in the software/API of the OAK-1 device (https://docs.luxonis.com/projects/api/en/latest/components/nodes/object_tracker/#supported-object-tracker-types and https://dlstreamer.github.io/dev_guide/object_tracking.html) that could lead to different results regarding tracking accuracy of the system.

We hope that it is acceptable for the reviewer that we could not show a more precise evaluation of the tracking accuracy of our system at this stage, as many factors influence the respective tracking accuracy depending on the settings during image capture and post-processing, together with the difficulty of calculating true/false positive tracklets and false negative tracklets with the data that we have available at the moment. As the camera trap software is still under continuous development, we plan to run more targeted experiments to assess the tracking accuracy at different settings more rigorously, also with various insect species and under field conditions.

To account for the missing calculation of the appropriate metrics regarding tracking accuracy in the presented experiment, we changed the phrasing in lines 35, 123, 211, 634 and 668-669. Additionally, we now explicitly mention this shortcoming in the Discussion section in lines 673-683 and give more information about the object tracker in lines 666-668.

(4) In the Introduction (Line 102), Methods (Line 191) and Discussion (Line 588) authors state that the “...the whole pipeline runs at ~12.5 fps (1080p HQ frame resolution), which is sufficient to reliably track most insects…”. However, the evaluations presented in the insect tracking section (Line 320) utilize 4K HQ frames. This inconsistency raises concerns about which resolution was used for the actual classification task, as the authors mention both 1080p and 4K image synchronization throughout the manuscript. Since image resolution can significantly impact classification accuracy, it's crucial to clarify this point. If the camera trap system and its classification model were indeed tested on 4K captured (and cropped) images, reporting the processing speed for 4K frames would be more accurate and reflect the true operational speed of the system. Additionally, a brief discussion on the trade-offs between speed and resolution, and how these choices affect the monitoring of different insect species, would be valuable for readers.

Authors response: Thank you for making us aware of the possibly confusing inconsistency of both mentioning 1080p and 4K HQ frame synchronization resulting in different pipeline speeds, which in turn can influence the object tracker accuracy. Most of the images in the dataset for classification model training were cropped from 1080p frames. For the field deployment of the five camera traps, we also used 1080p HQ frame synchronization, including the real-world dataset that was used to evaluate the generalization capability of the classification model. The reason why we used 4K HQ frame synchronization for the object tracking experiment was to test the tracking accuracy under difficult conditions (see also previous response to comment 3). We tried to clarify these points by adding more information in lines 119, 211-217, 274-275, 278, 358-360, 377-378 and 635-636.

(5) I appreciate the detailed hardware assembly instructions provided in the documentation. However, it's currently unclear how the individual components connect to each other. To address this, I suggest incorporating a simple hardware schematic (complimenting Figure 1) to visually illustrate the connections between each component.

Authors response: Thanks a lot for this nice comment and useful suggestion. We agree that it makes sense to add a simple hardware schematic that gives more information about the connections between the individual components. We included this figure as “S1_Fig” in the supporting information (figure caption in lines 977-978) and refer to it in the text in lines 142-144. To account for the new numbering of the figures in supporting information, we changed lines 342, 435, 531, 596, 600, 980, 984, 988, 991 and 996 accordingly.

(6) Using artificial flower visits as a proxy for insect counts may not be as accurate as direct flower observations. This is because insect visitation to artificial flowers can be influenced by various characteristics of the flowers themselves. Please include a brief discussion on how the choice of platform materials and colors could affect capture rates, thereby improving the reliability of this method.

Authors response: Thanks for this suggestion. We added some more information about this aspect in lines 622-625. In this paragraph we are already discussing in lines 626-627 how “An updated platform design could further increase the visual attraction for a wider range of insect taxa or for specific target groups.”. As this is ongoing research, it is currently still difficult to provide more specific details.

(7) Line 87-89: The detection rate and accuracy of a deep learning model depends on various factors including its architecture and quality of the training dataset. Hence, the use of deep learning-based models for insect detection can result in false negative detections leading to underestimating insect counts. Please briefly mention this in the introduction and further discuss how this drawback could be overcome in the Discussion section.

Authors response: We agree that the accuracy (especially Recall) of the deployed insect detection model significantly affects the reliability of the estimated insect activity/abundance. We now mention this more clearly in the Introduction in lines 89-90. In the Discussion section, we now call attention to this aspect in lines 659-665.

Minor Comments

Presentation Structure: Authors have presented this study in easy to understand clear language. However, the structure the manuscript presented was confusing to me. I would like to propose authors to consider restructuring the manuscripts Methods and Materials and Results sections. Here Methods and Materials sections would contain two subsections on Camera Trap and Insect Classification model and only contain methodology associated with it. The Results section (or Experiments and Results) sections would contain all the experimental evaluations including results of YOLO models, classification model, field deployment etc, divided among two subsections on Camera Trap and Insect Classification model.

Authors response: While we understand that the evaluation metrics of the YOLO detection models and the EfficientNet-B0 classification model could have also been shown in the Results section, we think that our presented work differs to similar research articles that focus more strongly on the performance of the presented models. We see our presented models more as a baseline or example for interested users to train their own models on custom datasets. We tried to make this fact clear e.g. in lines 104-105 and 617-619. For this reason, the presentation of the model metrics on the validation/test datasets fit better in the Materials and methods section under the “Software” subsection. For unexperienced readers this also significantly enhances understanding of the whole pipeline that includes several steps from on-device detection and tracking to metadata post-processing that all build on top of each other.

Line 24: Not all traditional monitoring methods (e.g. focal flower observations or quadrat observations, transect walks) may provide data with high taxonomic resolution.

Authors response: This is correct and the reason why we purposely wrote “[…] traditional monitoring methods are widely established and can provide data with a high taxonomic resolution.” in lines 23-25. We hope that this indicates enough that the taxonomic resolution also depends on the specific method that is used.

Line 68 and 71: Other research that uses motion detection for insect monitoring include “van der Voort, Genevieve E., et al. "Continuous video capture, and pollinia tracking, in Platanthera (Orchidaceae) reveal new insect visitors and potential pollinators." PeerJ 10 (2022): e13191.”, “Steen, Ronny. "Diel activity, frequency and visit duration of pollinators in focal plants: in situ automatic camera monitoring and data processing." Methods in Ecology and Evolution 8.2 (2017): 203-213.”

Authors response: Thank you for mentioning these interesting research articles that are also using motion detection for camera-based insect monitoring. While there are a number of studies that use similar approaches, we wanted to keep the references as concise as possible and therefore decided to mainly cite papers that also include some form of AI-based post-processing of the resulting image/video data. Pegoraro et al. 2020 (Ref. 18) gives a broad overview of existing camera systems for pollinator monitoring, including Steen 2017.

Line 106: Please provide an appropriate reference and data on the speed of the hoverfly species.

Authors response: Unfortunately, we could not find any research articles that explicitly measured flight speed of hoverflies. The only information on hoverfly speed we could find are the following excerpts from Rotheray & Gilbert 2011: The natural History of hoverflies (ISBN 978-0-9564692-1-2):

Page 32-33: “[…] their wings have been recorded beating at between 120 and 150 beats per second, resulting in a forward speed of between 3 to 4 metres per second. To put this into context, hawkmoths (Sphingidae, Lepidoptera) beat their wings at much slower speeds of between 50 and 90 beats per second, but they fly faster at more than 5 metres per second. On the other hand, midges (Ceratopogonidae, Diptera) can beat their wings at more than 1000 times per second, but only move forward at less than half-a-metre per second.”

Page 36: “Analysis of film of hovering E. balteatus has revealed how rapidly hoverflies can accelerate. From a hovering start, acceleration begins in the first couple of wingbeats. At eight wingbeats or 40 milliseconds later, the hoverfly have moved forward about 4 cm and in another 40 milliseconds is has moved between 1-1.5 metres.”

We believe that without making a more specific statement in our manuscript, a reference is not necessary in this context. Determination of flying speed can be relative to e.g. other flower-visiting species and in this regard, most hoverflies (including Episyrphus balteatus) often fly faster than other species such as beetles, butterflies and many bees.

Line 123: Is 91 Wh the combined capacity of the two batteries?

Authors response: We added “[…] combined capacity” in line 139.

Line 127: What are the dimensions of the platform?

Authors response: We added the dimensions of the big and small platform that we were using in line 147. We also added the dimensions for the small platform shown in Fig 1 A to the figure caption in line 155.

Line 131: I suggest including the component list and associated cost values also in the supporting materials. This is because the cost provided in the manuscript may change over time.

Authors response: As the costs for the proposed components are changing frequently, we only added the approximate total costs in the manuscript text in line 150: “[…] ~700 € […]”. We will update the costs in the component list at the documentation website from time to time to show the current costs to interested readers. We do not think that it is necessary to add a component list to the manuscript as components will also change in new iterations of the camera trap and will be updated accordingly on the documentation website.

Line 158: It is unclear on what types other homogeneous backgrounds the YOLO model was tested with. Could you please clarify?

Authors response: We made this clearer by adding “[…] (e.g. variations of the artificial flower platform design).” in lines 177-178.

Line 161: Why were the metrics not calculated for the test split?

Authors response: We also calculated the metrics for the test split. These were very similar compared to the metrics for the validation split for all YOLO models (e.g. for YOLOv5n mAP@0.5: 0.964, Precision: 0.966, Recall: 0.954). We report the metrics on the validation split as this is the default output while training the model and in line with similar research articles, e.g. Bjerge et al. 2023 (Ref. 45). By open-sourcing our datasets and Google Colab notebooks that were used for model training, everyone interested can reproduce the presented metrics together with the metrics for the test split.

Line 178: Please provide references for Kalman Filter and Hungarian Algorithm.

Authors response: We added two references for object tracking with the Kalman Filter and Hungarian Algorithm in lines 198 and 861-870. We renumbered the subsequent references accordingly.

Line 191: [See Major comment 3 and 4]

Authors response: Lines 211-212 were changed accordingly to major comments 3 and 4.

Line 199: Please include the type of metadata recorded in the figure caption.

Authors response: We included the most important metadata in the caption of Fig 2 in lines 224-225.

Line 205: Please include more information on how the power consumption was measured. What device did you use to measure the energy consumption? Under what ambient conditions (temperature, humidity) was the test conducted? Was the solar panel connected to the system during this test? Were 5 insects tracked simultaneously or sequentially during the test? Also, what was the camera resolution?

Authors response: Thanks for this suggestion. We included more details about how the power consumption was measured in lines 231-235 and 237-238.

Line 208: Was the estimate of 20 hours calculated considering the threshold value mentioned in line 210?

Authors response: We did not consider the threshold value for calculating the total estimate of 20 hours runtime, as this can differ depending on the user settings.

Line 229: Could you please explain why YOLOv5 was used on captured images first to classification without directly using EfficientNet-B0 on captured images?

Authors response: We used a modified Python script from our YOLOv5 fork that supports several classification model architectures, including EfficientNet-B0 (https://github.com/maxsitt/yolov5/blob/master/classify/predict.py). We made this clearer by adding more information in line 262.

Line 234: Please provide a reference to the EfficientNet-B0 model.

Authors response: Thanks for finding this missing reference when we first mentioned the EfficientNet-B0 model. As we now explicitly mention the model architectures that we compared in line 291 (see also response to comment 9 of reviewer #1), we changed line 267 to be more general on the different model weights that were all pre-trained on ImageNet. We included a reference at the new first mention of EfficientNet-B0 in line 291 and deleted the previous citation in line 294. We also added a reference for ResNet-50 in line 291 and 879-881. We renumbered the subsequent references accordingly.

Line 259: Could you please explain why high inference speed is critical for this step. As the images are classified offline and as the main aim is to achieve the best classification accuracy, shouldn’t the accuracy be prioritized over inference speed?

Authors response: Thank you for this reasonable question. It is correct that the accuracy is prioritized over inference speed due to the classification model running offline. The only reason why inference speed should also be considered, is that a higher inference speed usually lets models also run faster on suboptimal hardware (e.g. only CPU without GPU) that users might only have available. Also, the classification of large numbers of images will take less time with faster models. However, even if only accuracy is accounted for, the EfficientNet-B0 models were still superior to the YOLOv5-cls and ResNet-50 models (see S2 Table). To avoid confusion regarding this aspect, we changed lines 293-296 accordingly.

Line 313: Please change “mm” to millimeters.

Authors response: We changed “mm” to “millimeters” in line 350.

Line 324: Please provide the video settings used by the smartphone camera including its resolution and framerate. Also, could you please provide more detail on the speed the smartphone camera video was played at? (e.g 50% of original speed).

Authors response: We included the resolution and framerate of the videos that were filmed with the smartphone camera in line 363. We also included details about the reduced playback speed in line 364.

Line 364: Please explain why a threshold of 70% was used? Why not set a lower value allowing us to record more data?

Authors response: We use this threshold as a default (can be changed by the user) to optimize recording efficiency if less sunlight is available to charge the batteries and to avoid recording gaps. We added this explanation in lines 408-409 and shortened the whole sentence in line 410.

Line 416: I suggest authors include the S3 table in the manuscript as it reflects the performance of the classification model on the real world data. Also Table 2 can be moved to Supplementary materials.

Authors response: Table 2 shows the metrics of our classification model on the dataset test split, which is in line with similar research works and can be regarded as a standard form of reporting model performance. For these reasons, we would like to keep it in the manuscript. We additionally tested the classification model on a real-world dataset to demonstrate that model performance will most probably differ in the case of non-curated image data. However, with the limited scope of this additional test, the generalization capability of the classification model can only be estimated. The shown metrics will change for other real-world datasets and a much larger dataset including more images with a higher variety of captured insect species is needed to evaluate the generalization capability of the model more rigorously. We tried to make readers aware of this fact in the Discussion section, e.g. in lines 644-649 and 692-699. Therefore, we would like to keep S3 Table in the supporting information and only show the more comprehensive Fig 8 as estimation of the generalization capability of the classification model.

Line 516: I suggest authors present the analysis of hoverfly behavior under a subsection “Example data analysis" or " Case Study”.

Authors response: Thanks for the suggestion, but we think that the presentation of data with focus on captured hoverfly images is still a good fit for the subsection “Insect monitoring data”, as we not necessarily analyzed hoverfly behavior, but only show the numbers of unique hoverfly tracking IDs (= estimated activity/abundance) over the course of the field deployment of the camera traps.

Line 579: Could you please provide any references to support the statement that the artificial flower platform can be standardized similarly to yellow pan traps. [Also see Major comment 6].

Authors response: We do not know of any reference that would support the statement that the artificial flower platform can be standardized similar to yellow pan traps. However, we do not think that a reference is needed in this context, as this is only a general statement without specific details. As response to major comment 6, we added more information regarding this aspect in lines 622-625.

Line 582: Please change the “camera trap system” to “camera trap hardware” as the software system was not evaluated for monitoring insects visiting real flowers or in outdoor settings.

Authors response: Even though we did not evaluate the provided detection models and software on the use cases mentioned in lines 629-631, we believe that it will be possible to use the same hardware and software system also for monitoring of other insects on different backgrounds, including real flowers. This will require training of a new detection model with a dataset that reflects the new environment. In lines 659-665 we now explain how the required training dataset size increases with complexity of the system that should be monitored. For this reason, we kept “camera trap system”, but added information about the required training of a new/adapted detection model and changed the phrasing of the sentence in lines 628-629.

Line 583: I agree with authors that the presented hardware system can be used to monitor insects visiting real flowers. However, it is unclear how the software solution will translate to monitoring insect visits to real flowers. Basic monitoring insect visit to real flowers requires detecting insects in an image, obtaining its coordinates, tracking its position and movements with changes in environment and its posture, and comparing insect coordinates with flower coordinates to identify flower visits. Could you please mention how the presented methods can achieve these requirements, or alternatively present these requirements as future work. This could be an expansion to the discussion presented under Insect detection and tracking subsection in the Discussion section.

Authors response: We are actively working on adapting the camera trap system to monitor insects visiting real flowers at the moment. This requires not only adaptation of the detection model (as indicated in e.g. lines 628, 652-657 and 659-665) but also of the hardware setup (e.g. fixing of the flower at the right distance to the camera to avoid wind movements). In our opinion we already made it clear that a deployment of the camera trap system to new environments requires specific adaptations to achieve monitoring results with a sufficient accuracy. More detailed explanations on the specific requirements are out of scope for this work and will be presented in upcoming papers about updates of the camera trap system.

Line 603: Under insect detection and tracking section please include a discussion on how the methods presented in this study can be implemented with different IoT platforms and how development of more efficient computational platforms can leverage the results of this study.

Authors response: A substantial part of the software that we use on the camera trap hardware uses the DepthAI Python API (https://docs.luxonis.com/projects/api/en/latest/) that is specific for the Luxonis OAK devices and therefore difficult to implement with other IoT platforms. Our provided datasets for insect detection and classification model training can of course be used also for other purposes, but this should be clear for interested readers and does not have to be mentioned explicitly in the manuscript from our point of view. In the subsection “Future perspectives”, we already indicate that improved hardware platforms will make the presented approaches more attractive for a wider audience in lines 737-739.

Line 609, 611, 625: Studies cited in [45,46,48] can also be discussed in the introduction section.

Authors response: We wanted to keep the introduction as brief and concise as possible to make it easier to read and more interesting also for readers that are new to the field of automated insect monitoring. In our opinion these studies are a better fit in the Discussion section after the reader already got a grasp of the potential difficulties of automated visual insect monitoring.

Line 611: Other research that used motion enhancement includes “Ratnayake, Malika Nisal, Adrian G. Dyer, and Alan Dorin. "Tracking individual honeybees among wildflower clusters with computer vision-facilitated pollinator monitoring." Plos one 16.2 (2021): e0239504.”

Authors response: Thanks for the suggestion. We added Ratnayake et al. 2021 to the references in lines 659 and 913-915. We renumbered the subsequent references accordingly.

Line 620 - 622: Currently there is research being conducted on re-identification of insects (see. Borlinghaus, Parzival, Frederic Tausch, and Luca Rettenberger. "A Purely Visual Re-ID Approach for Bumblebees (Bombus terrestris)." Smart Agricultural Technology 3 (2023): 100135.”). Please discuss the possibility of using a similar mechanism for the proposed study to improve its sampling accuracy.

Authors response: Thanks for making us aware of this very interesting work. However, this approach still seems to be in an early stage and we can see many potential difficulties when trying to implement something similar for our proposed system (e.g. this will probably only work for a specific set of insect species). We think that this approach is out of scope for our manuscript but will keep a close watch on further developments in this field that could be implemented in the future.

Fig 2.: Please label the purple line from Script node to cropped detections.

Authors response: To keep the labeling consistent, we labeled the purple line (HQ frame) going into the script node in the new version of Fig 2. It should be clear that the colored lines going out of the script node represent the synchronized inputs.

Fig 4: Please indicate the start and end of the recording period.

Authors response: We included information about the start and end of the recording at the two power spikes in the figure caption for Fig 4 in lines 247-248.

Fig 6 and Fig 8: Please label the color bar.

Authors response: We do not think that it is necessary to label the color bar representing the proportion of images that were classified to a predicted class to the total number of images per true class for the normalized confusion matrix plots shown in Fig 6 and Fig 8. We have not seen a normalized confusion matrix with labeled color bar in any similar research works. To make the content of the cell labels clearer, we added an explanation in the figure caption for Fig 6 in lines 303-304 and for Fig 8 in lines 469-471.

Fig. 7: I suggest removing the “15 min recording” from axis titles to simplify the plot. Also rename y axis to “Ground truth” or “Manual video observations” and the x axis to “Camera trap recordings”.

Authors response: We agree that the suggested changes could simplify the plot axis labels of Fig 7. However, we think that our current labels describe the underlying data more precisely and could avoid potential misinterpretation compared to the suggested labels.

Fig 11. I suggest moving this figure to supplementary materials. Results presented in 11F are a bit confusing as all the camera traps were not deployed for the same period of time. I recommend removing plot 11F. Could you please provide more information on how sunshine was measured? Please adjust the secondary y axis scale to match that of the primary y axis. Also include a legend describing what each line in the plot represents.

Authors response: In our opinion, Fig 11 shows an important characteristic of the camera trap, namely the charge level of the PiJuice battery depending on the duration of sunshine available per day to charge the batteries. While it is correct that the plot in Fig 11 F does not give an overview of all camera traps during the whole period, as some were deployed later in the season, we still think that it gives valuable information to the reader without misleading the interpretation of the merged charge level data and mean of the sunshine duration at the deployment sites. For these reasons, we would like to keep Fig 11 F. We got official data from the German Meteorological Service (“Deutscher Wetterdienst”) on sunshine duration, measured at the nearest weather station for each deployment site. The only information that we could find on their website, is that the sunshine duration is measured with “opto-electronic sensors” (German source: https://www.dwd.de/DE/fachnutzer/landwirtschaft/dokumentationen/allgemein/basis_sonnenscheindauer_doku.html). As we are using a credible official source for this data, we do not think that a detailed explanation of how the sunshine was measured is required in this context. We purposely decreased the scale for the secondary y-axis to enhance the readability of the plots, as both the lines for charge level and sunshine duration would overlap if both axes would be scaled to the same size, which would significantly impair readability. As suggested, we added a legend to the new version of Fig 11 in plot Fig 11 A and in plot Fig 11 F, describing what each line is representing in the plots. To enhance readability, we also moved the y-axis label to Fig 11 A (previously Fig 11 C) and removed the y-axis label from Fig 11 E.

Fig.12: Please make all y axis scales 12A-12E same to enable easy comparison of data across camera traps.

Authors response: We agree that the same axis scale is important for many examples where the same kind of data should be compared between e.g. different sites/methods/times. However, we do not think that scaling all y-axes in Fig 12 A-E the same would enhance interpretability in this case. In lines 406-416, we make it clear that several factors influenced the image capture for each camera trap in our presented data. Especially the significant differences in the total recording time (shown in Table 3) make it impossible to directly compare the number of unique tracking IDs per class (= estimation of abundance/activity) between the camera traps. Especially for Fig 12, we believe that the distribution of the captured classes gives the most interesting information for the readers. And this distribution is better illustrated if the individual plots are not scaled to the same size, which would significantly impair readability regarding class distribution for all plots except Fig 12 B and Fig 12 F. For these reasons, we would like to keep the scaling of the y-axes in Fig 12.

Fig 14: As the recording time per day varies between camera traps, I suggest normalizing the number of unique tracking IDs by the recording time. Please make all y axis scales 14A-14E same to enable easy comparison of data across camera traps

Authors response: As the reviewer is suggesting, we already normalized the number of unique tracking IDs by the recording time for Fig 15 and explain this in lines 575-577. We still think it is interesting to see the recording time, shown on the secondary y-axis in Fig 14, in relation to the total number of unique tracking IDs for all camera traps. For similar reasons as already mentioned in the previous response to the comment on Fig 12, we think that scaling the y-axes the same is not necessarily required in this case. Enabling an easier readability for each individual camera trap regarding the peaks of captured hoverfly tracking IDs in Fig 14 is more important for the readers in our opinion. For this reason, we would like to keep the scaling of the y-axes in Fig 14.

Fig 15: As the discussion does not extensively analyze the relationship between rainfall and hoverfly activity, this Figure can be removed.

Authors response: As mentioned in the previous response to the comment on Fig 14, we show the normalized number of unique tracking IDs (per hour of recording) in Fig 15. We added the amount of rainfall in the secondary y-axis to include a potential influencing factor on hoverfly activity, that could be easily shown in the plot. We indicate the potential influence of rainfall on hoverfly activity in lines 584-585. For the scope of this work, we do not think that it is necessary to extensively analyze this potential relationship in the Discussion section.

---

## [Decision Letter · Decision Letter 1]

13 Feb 2024

PONE-D-23-38812R1Insect Detect: An open-source DIY camera trap for automated insect monitoringPLOS ONE

Dear Dr. Sittinger,

Thank you for submitting your manuscript to PLOS ONE. After careful consideration, we feel that it has merit but does not fully meet PLOS ONE’s publication criteria as it currently stands. Therefore, we invite you to submit a revised version of the manuscript that addresses the points raised during the review process.

We look forward to receiving your revised manuscript.

Kind regards,

Ramzi Mansour

Academic Editor

PLOS ONE

Reviewers' comments:

Reviewer's Responses to Questions

**Comments to the Author**

1. If the authors have adequately addressed your comments raised in a previous round of review and you feel that this manuscript is now acceptable for publication, you may indicate that here to bypass the “Comments to the Author” section, enter your conflict of interest statement in the “Confidential to Editor” section, and submit your "Accept" recommendation.

Reviewer #1: (No Response)

Reviewer #2: All comments have been addressed

2. Is the manuscript technically sound, and do the data support the conclusions?

Reviewer #1: Yes

Reviewer #2: Yes

3. Has the statistical analysis been performed appropriately and rigorously? 

Reviewer #1: Yes

Reviewer #2: N/A

4. Have the authors made all data underlying the findings in their manuscript fully available?

Reviewer #1: Yes

Reviewer #2: Yes

5. Is the manuscript presented in an intelligible fashion and written in standard English?

Reviewer #1: Yes

Reviewer #2: Yes

6. Review Comments to the Author

Reviewer #1: 

The paper titled "Insect Detect: An Open-Source DIY Camera Trap for Automated Insect Monitoring" represents a commendable effort, offering a hardware solution for insect counting in natural environments and their classification into taxonomic orders, thereby providing significant value. This contribution to the scientific community, encompassing both the hardware solution and the open-source code, underscores its importance. However, regrettably, the text itself presents challenges in readability. It's not the language per se, but rather the overall structure and organization of section headings that can be perplexing. In my view, the authors should strongly consider revising some section headings and streamlining certain portions of the paper to enhance its accessibility and, consequently, its impact as a valuable scientific resource. Below, I outline specific questions regarding these aspects.

• Figure 5 could benefit from improvement. It's unclear why there are x and y dimensions labeled for all images, especially considering they are of different sizes. Shouldn't the images be resized to a consistent size? Removing the axes would free up space, allowing for slightly larger images and thus improving visibility.

• In the paragraph beginning at line 285, you mention that the dataset is divided into 80% for training, 10% for validation, and 10% for testing. Could you clarify how this division is achieved? Is it done randomly or as 80%, 10%, 10% for each class?

• In the paragraph starting at line 285, you mention that you are utilizing the YOLOv5 classifier. Could you provide insights into any differences between training a typical ResNet50 versus a YOLOv5 ResNet50? Your commentary on this topic would be greatly appreciated.

• In the paragraph starting at line 285, you mention using an image size of 128x128 for insect classification and 320x320 for initial insect detection. While the results are promising at these resolutions, could you comment on whether using a larger image size would further improve results? Additionally, considering hardware constraints, especially if this is not being done on the Pi-module, how does this impact your choice of image size?

• I find the section headings confusing, particularly regarding the processing pipeline. Starting with section 2.2 on software, why is insect detection (2.2.1) not within the processing pipeline (2.2.2)? In my opinion, the software section should be renamed "Processing Pipeline," or better yet, remove the hardware and software distinctions entirely and replace them with more thematically fitting headings, such as 2.1. Pi-module - Insect Detect and 2.2. Processing Pipeline.

• In line 183, Table 1 there are results for insect detection in the methods section. Shouldn't these results be in the results section instead?

Reviewer #2: 

This paper introduces an automated, do-it-yourself camera trap system for studying flower-visiting insects. The system comprises two key components: a real-time camera equipped with a deep learning-based object detector. This detector identifies and captures images of insects landing on an artificial platform. Additionally, an insect classification model analyzes the captured images to identify the species of each insect.

The authors have adequately addressed the comments and concerns raised in the previous review, incorporating the necessary changes into the manuscript. I encourage the authors to continue updating and maintaining the documentation and software associated with this research, as it provides a valuable tool for researchers and citizen scientists engaged in insect monitoring studies.

7. PLOS authors have the option to publish the peer review history of their article (what does this mean?). If published, this will include your full peer review and any attached files.

Reviewer #1: No

Reviewer #2: No

---

## [Author Response · Author response to Decision Letter 1]

26 Feb 2024

Reviewer #1:

Comments to the Authors:

The paper titled "Insect Detect: An Open-Source DIY Camera Trap for Automated Insect Monitoring" represents a commendable effort, offering a hardware solution for insect counting in natural environments and their classification into taxonomic orders, thereby providing significant value. This contribution to the scientific community, encompassing both the hardware solution and the open-source code, underscores its importance. However, regrettably, the text itself presents challenges in readability. It's not the language per se, but rather the overall structure and organization of section headings that can be perplexing. In my view, the authors should strongly consider revising some section headings and streamlining certain portions of the paper to enhance its accessibility and, consequently, its impact as a valuable scientific resource. Below, I outline specific questions regarding these aspects.

1. Figure 5 could benefit from improvement. It's unclear why there are x and y dimensions labeled for all images, especially considering they are of different sizes. Shouldn't the images be resized to a consistent size? Removing the axes would free up space, allowing for slightly larger images and thus improving visibility.

Authors response: Thank you for your suggestion. The individual images in Fig 5 show an example of the respective class from the dataset for classification model training. It is correct that all images are resized to the same size during classification model training and inference (in our model 128x128 pixel). However, all images shown in Fig 5 were automatically captured with the camera trap by cropping the detections (bounding box area) from synchronized HQ frames, which results in different image sizes, depending on the size of the bounding box (e.g. smaller for “ant”, bigger for “lepi”). To improve the clarity and comprehensibility of Fig 5, we decided to resize all example images to the same dimension. To still give the readers the important information about the original image sizes (as captured by the camera trap), we added x- and y-axis labels representing the original pixel values. We believe that the importance of this information outweighs the improved visibility of slightly larger images without the axes.

2. In the paragraph beginning at line 285, you mention that the dataset is divided into 80% for training, 10% for validation, and 10% for testing. Could you clarify how this division is achieved? Is it done randomly or as 80%, 10%, 10% for each class?

Authors response: Thank you for finding this important missing information. We added “[…] randomly […]” in line 296 to make it clear that the dataset was split randomly. This still means that all images of each class are split into the train (70%), validation (20%) and test (10%) subsets with the same ratio. However, as we are dealing with an imbalanced dataset, this results in different numbers of images per class in each subset (see also Table 2 for the test dataset). As this information was also missing for the dataset for detection model training, we added “[…] randomly […]” also in line 182.

3. In the paragraph starting at line 285, you mention that you are utilizing the YOLOv5 classifier. Could you provide insights into any differences between training a typical ResNet50 versus a YOLOv5 ResNet50? Your commentary on this topic would be greatly appreciated.

Authors response: This is indeed an interesting point. In fact, we only use a slightly modified version of the Python script for classification model training from the YOLOv5 repository (https://github.com/maxsitt/yolov5/blob/master/classify/train.py) and not a specific “YOLOv5 classifier”. The ResNet50 and EfficientNet-B0 models are taken from the torchvision.models subpackage and were pre-trained on the ImageNet dataset (more info here: https://github.com/ultralytics/yolov5/tree/master#classification under “Classification Checkpoints”). In theory, the YOLOv5 script for classification model training supports all available PyTorch implementations of the (optionally pre-trained) classification model architectures from torchvision.models (see here: https://pytorch.org/vision/stable/models.html#classification). As a more thorough comparison of many different classification model architectures was out of scope for this manuscript, we only focused on some of the pre-trained weights published together with the YOLOv5 v6.2 release (https://github.com/ultralytics/yolov5/releases/v6.2). We are currently working on optimizing this classification model training and inference workflow by comparing more architectures. Results from these tests will be published in an upcoming paper. To come back to the reviewer’s question: the ResNet50 and EfficientNet-B0 are the official PyTorch implementations of both model architectures, available in the torchvision package and pre-trained on the ImageNet dataset. To avoid overcomplicating this topic, we are not including this additional information in the manuscript as it would probably only confuse most of the (non-professional) readers and would not add important information to our already described workflow.

4. In the paragraph starting at line 285, you mention using an image size of 128x128 for insect classification and 320x320 for initial insect detection. While the results are promising at these resolutions, could you comment on whether using a larger image size would further improve results? Additionally, considering hardware constraints, especially if this is not being done on the Pi-module, how does this impact your choice of image size?

Authors response: Thank you for this question. First, regarding the resolution of 320x320 pixel for insect detection (and tracking) running on the device in real time: We explain in lines 203-205 that this downscaled resolution (LQ frames) increases the inference speed of the detection model and thereby the accuracy of the object tracker. We tested different resolutions for our use case and 320x320 pixel resulted in the optimal balance between accuracy and speed. Depending on the specific use case (background, distance from camera to object, object size, object speed etc.) the optimal resolution could be lower or higher than what we are using. We mention multiple times in the Introduction and Discussion that users can train models on their custom datasets that are more adapted to their specific use case. By providing detailed documentation and Google Colab notebooks we enable also non-professional users to train their custom models, optionally on a different image resolution that is better suited for their use case (e.g. even lower resolution to increase the model’s inference speed). Second, regarding the resolution of 128x128 pixel for insect classification: we tested different image resolutions (see also S2 Table) for classification model training/inference. Resizing the insect images (= cropped detections) to a resolution of 128x128 pixel means that most of the images in our training dataset were upscaled. This led to better results compared to downscaling more of the images. In the Google Colab notebook for classification model training, we give users the option to calculate the metrics of their custom image dataset (https://colab.research.google.com/github/maxsitt/insect-detect-ml/blob/main/notebooks/YOLOv5_classification_training.ipynb#scrollTo=aVZgGZtc8skH). We recommend using the 90th percentile of the image sizes (divisible by 32) as reference point to set the image size for model training (and later inference). However, this is only an estimated reference point and it is still required to test different resolutions to find the optimal model when training on a new custom dataset. A different aspect is the HQ frame resolution that is used for cropping the detections. We explain in lines 217-225 that a higher resolution can increase classification accuracy but can also decrease tracking accuracy as the inference speed of the detection model is reduced.

5. I find the section headings confusing, particularly regarding the processing pipeline. Starting with section 2.2 on software, why is insect detection (2.2.1) not within the processing pipeline (2.2.2)? In my opinion, the software section should be renamed "Processing Pipeline," or better yet, remove the hardware and software distinctions entirely and replace them with more thematically fitting headings, such as 2.1. Pimodule - Insect Detect and 2.2. Processing Pipeline.

Authors response: Thank you for your suggestion. We built the entire workflow (from on-device detection to metadata post-processing) in a modular way, so that users can change their specific settings or deployed models at different stages of this workflow, without affecting previous/subsequent steps. For example, own custom models can be used for the insect detection without the need to change the on-device processing pipeline. Also, different classification models/workflows can be used for insect images captured with the camera trap. It is also possible to use the classification model/workflow on other images that were not necessarily captured with our proposed camera trap system. To indicate these different “workflow modules”, we used a subsection header for each step. We added “On-device […]” to the subsection headings in line 167 and 201 to clarify the distinction to the subsequent classification and metadata post-processing that is not happening on the camera trap.

6. In line 183, Table 1 there are results for insect detection in the methods section. Shouldn't these results be in the results section instead?

Authors response: Thank you for your suggestion. While the metrics of the detection models (and the classification model) are more often shown in the Results section in similar research articles, we focus more on non-professional readers and best possible clarity and accessibility. By already showing the model metrics in the respective subsections of the Materials and methods section, potential users are guided through the whole workflow in a more logical and cohesive way, without having to jump between sections. From our point of view, it is also easier to understand the subsequent processing steps if information about the model performance is given before explaining the on-device processing pipeline in more detail. In contrast to similar research works, we do not focus so strongly on the performance of the presented models and see them more as a baseline or example for interested user to train their own models on custom datasets. While we agree that this is probably a more unconventional way to present model metrics, we would like to keep them in the Materials and methods section for the mentioned reasons of increased accessibility.

Reviewer #2:

This paper introduces an automated, do-it-yourself camera trap system for studying flower-visiting insects. The system comprises two key components: a real-time camera equipped with a deep learning-based object detector. This detector identifies and captures images of insects landing on an artificial platform. Additionally, an insect classification model analyzes the captured images to identify the species of each insect.

The authors have adequately addressed the comments and concerns raised in the previous review, incorporating the necessary changes into the manuscript. I encourage the authors to continue updating and maintaining the documentation and software associated with this research, as it provides a valuable tool for researchers and citizen scientists engaged in insect monitoring studies.

Authors response: Thank you for your comment! We will continuously update and maintain the software and documentation to implement new features, further improve the detection and classification models and keep all requirements up to date.

---

## [Editor Report · Decision Letter 2]

29 Feb 2024

Insect Detect: An open-source DIY camera trap for automated insect monitoring

PONE-D-23-38812R2

Dear Dr. Sittinger,

We’re pleased to inform you that your manuscript has been judged scientifically suitable for publication and will be formally accepted for publication once it meets all outstanding technical requirements.

Kind regards,

Ramzi Mansour

Academic Editor

PLOS ONE

---

## [Editor Report · Acceptance letter]

12 Mar 2024

PONE-D-23-38812R2 

PLOS ONE

Dear Dr. Sittinger, 

I'm pleased to inform you that your manuscript has been deemed suitable for publication in PLOS ONE. Congratulations! Your manuscript is now being handed over to our production team.

Kind regards, 

on behalf of

Dr. Ramzi Mansour 

Academic Editor

PLOS ONE